# HDAC Inhibitors: Innovative Strategies for Their Design and Applications

**DOI:** 10.3390/molecules27030715

**Published:** 2022-01-21

**Authors:** Mateusz Daśko, Beatriz de Pascual-Teresa, Irene Ortín, Ana Ramos

**Affiliations:** 1Department of Inorganic Chemistry, Faculty of Chemistry, Gdańsk University of Technology, Narutowicza 11/12, 80-233 Gdańsk, Poland; mateusz.dasko@pg.edu.pl; 2Departamento de Química y Bioquímica, Facultad de Farmacia, Universidad San Pablo-CEU, CEU Universities, Urbanización Montepríncipe, 28925 Alcorcón, Spain; bpaster@ceu.es

**Keywords:** histone deacetylases (HDACs), PROTACs, folate conjugates, positron emission tomography (PET), fluorescent probes, nanoparticles, dendrimers

## Abstract

Histone deacetylases (HDACs) are a large family of epigenetic metalloenzymes that are involved in gene transcription and regulation, cell proliferation, differentiation, migration, and death, as well as angiogenesis. Particularly, disorders of the HDACs expression are linked to the development of many types of cancer and neurodegenerative diseases, making them interesting molecular targets for the design of new efficient drugs and imaging agents that facilitate an early diagnosis of these diseases. Thus, their selective inhibition or degradation are the basis for new therapies. This is supported by the fact that many HDAC inhibitors (HDACis) are currently under clinical research for cancer therapy, and the Food and Drug Administration (FDA) has already approved some of them. In this review, we will focus on the recent advances and latest discoveries of innovative strategies in the development and applications of compounds that demonstrate inhibitory or degradation activity against HDACs, such as PROteolysis-TArgeting Chimeras (PROTACs), tumor-targeted HDACis (e.g., folate conjugates and nanoparticles), and imaging probes (positron emission tomography (PET) and fluorescent ligands).

## 1. Introduction

Epigenetic mechanisms have a central role in the control of biological processes. Histone deacetylases (HDACs) belong to the machinery of the epigenetic apparatus and play a crucial role in the formation of corepressor complexes involved in chromatin remodeling and gene expression. On the contrary, there are histone acetyltransferases (HATs) that act as coactivators. A good balance between the opposing actions of HATs and HDACs allow the epigenetic regulation of gene expression.

The repression process mediated by HDACs starts by the action of protein readers, such as methyl-binding proteins (MBPs), which bind methylated DNA, recruiting HDACs. Then, HDACs deacetylate the ε-NH_2_ groups of the lysine residues on histone tails. This deacetylation also tightens the interaction between positive charged histones and the negative charged DNA, leading to chromatin compaction, and therefore, inducing transcriptional repression through chromatin condensation [1,2]. On the other hand, HATs transfer acetyl groups to amino-terminal lysine residues in histone, causing chromatin expansion, increasing the accessibility of regulatory proteins to DNA, and allowing transcription (Figure 1) [3,4]. Thus, any chromatin alteration, due to improper HATs targeting, HDACs overexpression, or epigenetic changes in DNA methylation, can lead to the emergence and evolution of a wide range of diseases [5]. Due to this, HDACs have become an important target for the treatment of several diseases and, therefore, the development of new HDAC inhibitors is on the rise [1,6,7].

Currently there are 18 types of HDACs, classified into two groups, which are also subdivided into four classes, according to their location, their homology, their enzymatic activity, their order of discovery, and their histone substrate specificity: the first group contains zinc-containing HDACs, which share a similar catalytic core for acetyl-lysine hydrolysis. This group comprises classes I (HDAC1, -2, -3, and -8), II (IIa: HDAC4, -5, -7, and -9; IIb: HDAC6 and -10), and IV (HDAC11). The second group consists of NAD-dependent HDACs (class III, also known as sirtuins, SIRTs 1–7), which need a nicotinamide adenine dinucleotide for the optimum use of their enzymatic activity [5].

To date, many selective and multitargeting HDAC inhibitors (HDACis) have been developed, and some of them have been approved for the treatment of cancer [8,9,10]. Most of the known HDAC isoforms demonstrate a highly conserved nature, possessing a Zn^2+^ ion located in their active site. Therefore, many of the developed HDACis are formed by a zinc binding group (ZBG) that interacts with the Zn^2+^ ion in the catalytic region and a “cap group” occupying the entrance to the active site. The ZBG and the “cap group” are typically connected through a hydrophobic linker. In most of the known HDACis, the ZBG are hydroxamic acid, anilide, or thiol (usually masked as a disulfide bond), which provide strong chelating properties of the catalytic Zn^2+^ ion. The general structure of these HDACis is represented in Figure 2.

Although the traditional drug design based on small molecules, such as HDACis, remains a powerful strategy for the development of novel therapies, these face major problems, such as drug resistance, especially in advanced diseases. This is mainly due to the alterations in the target, ineffective apoptosis, or activation of different pathways, among others. Furthermore, in the case of cancer, the lack of active targeting to tumor cells from conventional chemotherapy hinders the candidates’ chances of success in the clinic. Thus, this review aims to provide an overview of the progress in the field of innovative strategies based on HDACis, such as PROteolysis-TArgeting Chimeras (PROTACs), tumor-targeted HDACis (e.g., folate conjugates and nanoparticles), and imaging probes (positron emission tomography (PET) and fluorescent ligands). We have focused on the period from 2015 to present, except for folate conjugates and fluorescent ligands, for which we have included the reported examples since 2007, as they have not been reviewed previously.

## 2. PROteolysis-TArgeting Chimeras (PROTACs)

The use of PROTACs has become a promising approach to overcome resistance since they degrade instead of inhibit the target, with the advantage of reducing the systemic drug exposure and to counteract the protein expression that often accompanies inhibition of protein function [11]. This approach uses bivalent molecules that possess a Protein of Interest (POI) recruiter linked to an E3-ligase hijacker for selective protein degradation using the cell machinery. The general structure of PROTACs and their putative mechanism of action are shown in Figure 3. To function properly, cells need to discard excess or damaged proteins. The ubiquitin-proteasome system is a well-tuned cellular protein-disposal system which consists of a wide number of proteins that begin their activity with the ubiquitin-tagging of the POI to be degraded [12]. Ubiquitin (U) is a small structurally conserved protein that is ubiquitously present in all eukaryotic cells, and later gave its name to the ubiquitin protein family, which is made up of several members that share a highly conserved structure [13]. The ubiquitination process is ATP-dependent. The first step is the formation of the ubiquitin-adenylate, which is then transferred to a cysteine side chain of the ubiquitin activating enzyme E1 [14]. E1 transfers the ubiquitin to a highly conserved cysteine residue of the ubiquitin conjugating enzyme E2. E2 then selectively interacts with E3 ubiquitin-protein ligase or E3-ligase, which is responsible of recruiting the POI, favoring its ubiquitin labeling by proximity [15]. Once the POI is ubiquitinated, it is recognized by the proteasome. Finally, the tagged protein is unwound and passes through the central channel of the proteasome, which contains degrading proteases.

Since PROTACs also enable the degradation of undruggable proteins, which are not accessible to traditional small-molecule inhibitors, nowadays, the design and synthesis of PROTACs is being intensively investigated as a promising strategy, mainly in the development of anticancer agents [16]. More recently, the use of PROTACs is expanding to the treatment of immune disorders, viral infections, and neurodegenerative diseases [17].

The first generation of PROTACs was based on peptides as the E3-ligase recruiters. Due to their clinical limitations, mainly related to the metabolic instability and poor pharmacokinetic properties of peptides, new PROTACs that use different types of small molecules as the E3-ligase recruiters were designed: (1) *Nutlin-3a* **1** as a ligand of MDM2 (Mouse Double Minute 2 homologue); (2) *Bestatin* **2** as a ligand of cIAP1 (cellular inhibitor of apoptosis protein 1); (3) *Thalidomide* **3** and its derivatives *Lenalidomide* **4a** and *Pomalidomide* **4b,** as cereblon (CRBN) ligands; and (4) *MDK7525* **5** as von Hippel-Lindau (VHL) ligands, among others (Figure 4). These compounds provided improved ADME (Absorption, Distribution, Metabolism, and Excretion) properties to the corresponding PROTACs [11,18,19].

The development of PROTACs targeting HDACs is a new drug discovery strategy and only a few of these compounds have been developed so far [20,21,22,23,24,25]. One of the first examples was described by Schiedel et al. They developed a PROTAC molecule capable of degrading sirtuins (Sirts), a class of lysine deacetylases (KDACs), which were initially described as a class III HDAC [26]. The design and synthesis of compound **6** (Figure 5) was based on a combination of the sirtuin ligand and the E3 ligase recruiter. *Thalidomide*, through a linker containing a triazole group. A highly efficient click-reaction was employed for this purpose. Gratefully, compound **6** turned out to be a very potent and selective Sirt2 inhibitor with a IC_50_ value of 0.25 µM (the inhibitory activities against Sirt1 and Sirt3 were detected at concentrations higher than 100 µM). Western blot analysis after incubation of HeLa cells with compound **6** proved its ability to induce the degradation of Sirt2 in a dose- and time-dependent manner. A concentration-dependent effect was observed in the range of 0.05 to 5 µM, while at higher concentrations, the efficacy was limited. A maximum effect of the Sirt2 degradation was achieved after 2 h of treatment of HeLa cells.

In 2018, Yang et al. developed the first-in-class small molecule degraders for non-sirtuin HDACs by conjugating an HDACi with E3 ubiquitin ligase ligands [27]. The core moiety was based on *Nexturastat A*, a non-selective HDACi, which was connected to *Pomalidomide* as CRBN E3 ligase recruiter through different poly(ethylene glycol) (PEG)-triazole linkers. Unexpectedly, the initial studies indicated that obtained compounds were able to selectively degrade HDAC6 in MCF-7 cells. The most potent compound, **7** (Figure 5), was selected for further investigation. More detailed studies indicated that the degradation of HDAC6 by compound **7** occurred in a dose-dependent manner with the maximal effect between 0.123 µM and 370 nM. Half-maximal degradation (DC_50_) was achieved at 0.034 µM, and the maximum percentage of degradation (D_max_) was 70.5%. Significant degradation of HDAC6 occurred at around 2 h after treating MCF-7 cells with 2 µM of **7**. Furthermore, the dual effect of compound **7** was also proved: inhibition of HDACs and selective degradation of HDAC6 mediated via ubiquitin-proteasome for the targeted protein degradation (TPD). As the authors indicated, the selective degradation of HDAC6 was unexpected, but not surprising. It is known that the conjugation of non-selective kinase inhibitors or bromodomain proteins (BRDs) with a E3 ligand of ubiquitin ligase may lead to selective degraders [28,29].

Another series of new PROTACs targeting HDAC6 was synthesized by An et al. in 2019 [30]. In the design process, *Nexturastat A* (a selective HDAC6 inhibitor) was selected as the target-interacting moiety. *Nexturastat A* was connected to *Pomalidomide* as CRBN ligand using linkers of different lengths. The most efficient degradation of HDAC6 was achieved in the presence of compound **8** (Figure 5). The effective degradation in different cell lines (most significantly in multiple myeloma (MM) cells) appeared at 100 nmol/L. The induced degradation of HDAC6 occurred rapidly and specifically. Furthermore, it was dependent on the proteasome activity. The comparable growth inhibitory activities of HDAC6 degrader **8** and *Nexturastat A* on myeloma cells put forward its application as a possible alternative treatment strategy. More detailed in vivo studies are currently in progress.

In 2019, the same research group developed other series of *Nexturastat A*-based PROTACs targeting HDAC6 [31], where linkers bearing *Pomalidomide* were connected to the terminal aromatic ring of *Nexturastat A* via amide coupling. All compounds could induce degradation of HDAC6; however, compound **9** (Figure 6) achieved the best degradation of the protein in a wide range of cell lines (e.g., HeLa, Mino, Jeko-1, HUVEC, MM.1S, and MDA-MB-231). HDAC6 was degraded even at 1 nM concentration of compound **9** after incubation in MM.1S cell lines, similarly to PROTAC **8**, which was used as reference. Incubation of MM.1S cells with compound **9** (DC_50_ value of 3.2 nM) led to apparent HDAC6 degradation within 1 h and reached maximum degradation at around 6–8 h. Moreover, HDAC1, HDAC2, and HDAC4 were not degraded by this compound.

*Nexturastat A* was also used as a target-interacting moiety in the design strategy of a wide series of HDAC6 targeting PROTACs by Wu et al. [32]. A great number of new compounds containing *Pomalidomide* as CRBN ligand, connected through linkers of different lengths to the terminal aromatic ring of *Nexturastat A,* were synthesized. Both C4 and C5 of *Pomalidomide* were evaluated as possible positions for the conjugation with the target-interacting moiety. All compounds induced significant HDAC6 degradation, with **10** (Figure 6) reaching the highest activity. This compound was able to induce 82.7 and 74.9% degradation of HDAC6 at 100 and 10 nM concentrations, respectively, and presented DC_50_ and D_max_ values of 1.65 nM and 86.26%, respectively. Compound **10** was found to be selective against HDAC6. The mechanism of action involved inhibition of HDAC6 by the *Nexturastat A* motif, degradation of the ikaros family of zinc fingers (IKZFs) by the *Pomalidomide* moiety, and degradation of HDAC6 through the formation of the ternary complex. Since compound **10** showed an EC_50_ value of 74.9 nM against MM cells, this compound could be considered as a novel therapy against multiple myeloma. To evaluate the binding affinity of novel CRBN ligands, the researchers developed a practical and efficient cellular assay [33]. Originally, the assay used MM.1S cells; however, since it does not involve any genetic engineering, it is relatively easy to transfer from one cell type to others. Based on the obtained results, several CRBN ligands were selected to prepare novel HDAC6 degraders. As an example, compound **11** (Figure 7), with a DC_50_ value of 1.9 nM, induced a significant degradation of HDAC6 at 3 nM concentration in MM.1S cells. Neither HDAC4 nor IKZs were targeted by compound **11**.

In 2020, the same research group developed new cell-permeable degraders employing VHL E3 ligase. Analogously to the previously obtained PROTAC **10**, VHL ligand was conjugated with *Nexturastat A*. All these compounds exhibited selectivity against HDAC6 [34], the most potent being **12** (Figure 7) with DC_50_ values of 7.1 nM and 4.3 nM and DC_max_ values of 90% and 57% in human MM.1S and mouse 4935 cell lines, respectively. At the highest concentration for the degradation of HDAC6, 100 nM, **12** did not induce degradation of other HDACs or IKZFs. Mechanism studies proved that compound **12** mediates degradation of HDAC6 via proteasome. The decrease of HDAC levels started at 30 min and reached maximum values after 4 h.

In 2020, Smalley et al. synthesized compounds **13a**–**d** (Figure 8), which were able to degrade class I HDACs 1–3 [35]. CI-994 (**14**), a benzamide-based HDACi, was chosen as the target-interacting moiety. CI-994 was merged with two different E3 recruiters: a VHL ligand and a CRBN ligand. Since the linker length can play an essential role in degradation, six carbon and twelve carbon alkyl linkers were utilized in the design process. Fluorescent deacetylase in vitro assay with a purified ternary LSD1-CoREST-HDAC1 complex proved HDAC1 inhibitory properties of all the synthesized compounds. Those with C6-linker, **13a** and **13c,** showed significantly higher activities with values comparable to CI-994. Unexpectedly, incubation of E14 mouse embryonic stem cells, with compounds **13a**–**d**, for 24 h showed an increase of H3K56 acetylation with the less potent in vitro inhibition for **13b** and **13d**. Deeper studies in a human colon cancer cell line HCT116 with **13b** and **13d** indicated that they can degrade HDACs 1–3 in a dose-dependent manner, with the VHL-based ligand **13d** being more effective than the CRBN-based ligand **13b**. At 10 µM concentration of **13d**, HDAC1 and -2 underwent almost complete degradation, while HDAC3 did so to a lesser extent. It was also observed that compound **13d** induced death of HCT116 cells at similar levels to CI-994.

Cao et al., in 2020, developed another series of PROTACs targeting class I HDACs 1–3 based on CI-994 and its derivative **15** [36]. These selective HDACs degraders incorporate *Pomalidomide* as E3 ligase recruiter. Compound **16** (Figure 8) showed the most favorable properties in this series. The presence of the fluorine atom in the *o*-aminoanilide fragment of **16** provides selectivity for HDAC3 degradation. Western blot studies with RAW 264.7 cells showed that **16** was able to degrade HDAC3 with a DC_50_ value of 0.32 µM. The highest degradation of HDAC3 by **16** in RAW 264.7 cells was achieved after 6 h, and lasted at least 48 h. Under these conditions, no significant degradation of HDAC1 and HDAC2 was observed.

Roatsch et al. reported a new series of PROTACs capable of selectively degrading class I HDACs 1–3 [37]. The design and synthesis of these compounds were based on a *click* reaction coupling between macrocyclic tetrapeptides as HDACs inhibitors and *Thalidomide* as the CRBN ligand. All the synthesized compounds exhibited selective degradation of HDACs 1–3 in HEK293T cells in a time- and concentration-dependent manner without showing cytotoxic effects. Compounds containing a longer tyrosine-based spacer, were the most promising candidates. Specifically, **17** (Figure 9) induced efficient degradation of HDACs after two to four hours of treatment, with optimal HDACs 1–3 degradation at the 100–300 nM concentrations when tested in cell lysates. These favorable properties of **17** led the authors to propose these cyclic peptides as effective target-binding elements for PROTACs in general.

Xiao et al. designed and synthesized selective HDAC3 degraders based on benzoylhydrazide inhibitors to recruit both CRBN and VHL E3 ligases [38]. Preliminary biological evaluation indicated that PROTACs containing the VHL-recruiting moiety were more potent and selective for the degradation of HDAC3. One of such compounds, **18** (Figure 10), was selected for further evaluation as the most promising candidate. In vitro studies in MDA-MB-467 cells proved that the HDAC3 degradation by **18** occurred in a dose-dependent manner with a DC_50_ value of 42 nM after 14 h of treatment. Interestingly, no significant changes in the levels of HDAC1, HDAC2, and HDAC6 were observed. This degradation also occurred in a time-dependent manner and 70% of HDAC3 was eliminated after treatment for 8 h. Interestingly, this effect is long-lasting and reversible, as the steady level of HDAC3 rebounded 12 h after removal of **18** from the culture. Furthermore, compound **18** showed potent antiproliferative activity against several cancer cell lines, such as T47D, HCC-1143, MDA-MB-468, and BT549.

In 2020, Sinatra et al. developed an efficient solid-phase synthesis employing hydroxamic acids immobilized on resins (HAIRs) [39]. This strategy allowed the quick preparation of novel dual-target epigenetic and cytotoxic compounds. One of the compounds achieved using this method was PROTAC **19** (Figure 10), which was a conjugate of *SAHA* and *Thalidomide* connected via a PEG linker. Compound **19** was a potent pan-HDAC inhibitor and HDAC6 and HDAC1 degrader in HL60 cell line in a concentration-dependent manner. Treatment of HL60 cells with **19** led to a significant hyperacetylation of histone H3 and α-tubulin.

In 2020, Cao et al. synthesized three conjugates of different hydroxamic acids and *Bestatin*, a cIAP1 ligand, as HDACs degraders [40]. The most promising properties were exhibited by *Bestatin*-*SAHA* conjugate **20** (Figure 10), which presented higher potency than *SAHA* against several HDACs (i.e., HDAC1, HDAC6, and HDAC8). As an example, IC_50_ values of 30 and 63.5 nM against HDAC1 were reported by compound **20** and *SAHA*, respectively. Although compound **20** could not induce intracellular degradation of HDACs after 6 h of treatment, it could significantly decrease the intracellular levels of HDAC1, HDAC6, and HDAC8 after 24 h of treatment in a dose-dependent manner. Surprisingly, when 5 µM of *SAHA* were used, the downregulation of applied HDACs was also observed, indicating that this phenomenon was not due to proteasome degradation. To prove these findings, a study was conducted with a proteasome inhibitor. Interestingly, these derivatives showed very potent aminopeptidase N (APN) inhibitory properties and present interest as novel APN-HDAC dual inhibitors.

Summarizing, to date, several PROTACs targeting different HDACs have been reported. This new class of degraders exhibits high potency and selectivity for the degradation of HDACs; however, their therapeutic potential has not been widely applied. There is still a need for new classes of PROTACs targeting HDACs, with improved pharmacokinetic properties. Bioavailability of this type of compounds is one of the challenges that must be addressed to achieve compounds capable of reaching in vivo experiments and clinical trials. Thus, further investigation of such an innovative approach is of the utmost importance for medicinal chemistry and may be crucial in the development of new and more effective treatments for many diseases.

## 3. Tumor-Targeted HDAC Inhibitors

Traditional drug design strategies based on conventional inhibitors have demonstrated some weaknesses, such as toxicity, which is mainly caused by the non-specific delivery of the drug. Such problems often occur in anticancer drugs, which should specifically target cancer cells and be relatively harmless for normal cells. For this reason, several novel strategies aimed at developing more effective and specific anticancer therapies have been extensively investigated in recent years. One of these strategies focuses on targeting the therapeutic agent (including HDACis [41,42,43,44]) to tumor cells, through conjugation to dendrimers, nanoparticles, or a tumor-cell-specific ligand such as folic acid or peptides, among others; thus, reducing delivery to normal cells and its toxicity [45]. 

### 3.1. Folate-Based Tumor-Targeted HDAC Inhibitors

Since folic acid is an important cofactor in the one-carbon metabolism and is involved in the biosynthesis of essential components of nucleic acids [46], it is considered as one of the specific ligands for tumor cells. Folic acid must be completely supplied with the diet because mammalian cells are not able to synthesize it. The high hydrophilicity of the charged folate molecule hinders its passive diffusion across cell membranes and folate uptake must occur through other diverse transport systems, including the binding to the folate receptor (FR), which occurs with high affinity. Among the three subtypes of the folate receptor (FRα, FRβ, and FRγ), FRα is the most widely expressed. All of them are glycoproteins rich in cysteine residues that mediate the uptake of folates through an endocytosis process [46]. Once delivered inside the cell, the release of folates from FR is stimulated by the acidic environment of the endosome. Finally, the released folates are transported into the cytoplasm via proton-coupled folate transporter (PCFT) [47]. While the level of FRα in normal tissues is very low, higher expression levels are observed in numerous types of cancers [48,49,50]. This higher expression is directly associated with the higher folate demand of rapidly dividing cancer cells. Therefore, the folate dependency of many tumors has been therapeutic and diagnostically applied and FRα has been considered as a promising molecular target for the delivery of many cancer treatments. Examples of these applications are anti-FRα antibodies, high-affinity antifolates, folate-based imaging agents, and folate analogues or folate conjugates of drugs and toxins [51]. The general structure of tumor-specific conjugates and their uptake mechanism and action are shown in Figure 11.

In 2007, Suzuki et al. synthesized FR-targeted prodrugs of thiolate HDAC inhibitor *NCH-31* bearing a cleavable disulfide bond [52]. Among the synthesized conjugates, compound **21** (Figure 12) showed the highest HDAC inhibitory activity with an IC_50_ value of 0.27 µM in an HDAC fluorescent assay under reductive conditions. Cells studies in MCF-7 cell line showed a dose-dependent cell growth inhibitory activity of **21**. Importantly, a competition experiment with free folic acid led to a significant reduction of the growth inhibitory activity, indicating that the folate uptake mechanism is responsible for the cellular absorption of compound **21**. In addition, an accumulation of acylated histone H4 was observed, indicating that the cell growth inhibition correlates with the inhibition of HDACs.

In 2011, Carrasco et al. developed a new series of hydroxamate derivatives of folic acid and methotrexate [53]. All synthesized compounds showed good inhibitory activity against HDACs. A fluorometric assay with recombinant human HDAC8 and crude nuclear extract from HeLa cells showed that the most active derivative **22b** (Figure 12) was able to inhibit HDACs with IC_50_ values of 6.6 and 0.88 µM, respectively. Importantly, an enzymatic assay with dihydrofolate reductase (DHFR) isolated from *L. casei* demonstrated that derivative **22a** (Figure 12) possess DHFR inhibitory properties with an IC_50_ value of 0.25 µM. This is interesting in cancer, since DHFR induces the conversion of 7,8-dihydrofolate (DHF) to 5,6,7,8-tetrahydrofolate (THF), which participates in subsequent metabolic reactions, such as biosynthesis of thymidylate and purine nucleotide and, hence, DNA. Moreover, all compounds showed poor antiproliferative properties against A549 (non-small cell lung carcinoma) and PC-3 (human prostate cancer) cell lines.

Recently, in 2015, Sodji et al. synthesized tumor-targeting compounds, where histone deacetylase inhibitors were conjugated to pteroic compounds and folic acids, respectively [54]. HDAC1, HDAC6, and HDAC8 inhibitory activities were evaluated for all the synthesized derivatives. Although compound containing the trimethylene chain demonstrated significantly weaker HDAC inhibitory properties than other pteroic acid analogs, this compound resulted a selective HDAC6 inhibitor. Subsequent elongation of the linker improved HDAC1 and HDAC6 inhibitory effects. Among all synthesized compounds, the most potent inhibitors were **23** with an HDAC1 IC_50_ value of 16.1 nM and **24** with an HDAC6 IC_50_ of 10.2 nM (Figure 12). Folate derivatives were generally less potent than the pteroic inhibitors. The final evaluation of anticancer activity in HeLa (cervical carcinoma) and KB (oral carcinoma) cells indicated that only two pteroic hydroxymates were active in the micromolar range.

### 3.2. Dendrimers-Based Tumor-Targeted HDAC Inhibitors

Dendrimers are synthetic and usually symmetric, spherical compounds, which have found application in medicinal chemistry, including drug delivery, tissue engineering, and gene transfection. Due to their controllable chemical topology and branched structure, dendrimers have been widely used as ideal drug delivery systems. Their narrow polydispersity and nanometer sizes allow them to pass across biological barriers. It is important to note that the host of the ligand in the dendrimer structure can appear inside (by endoreceptors) or at the periphery (by exoreceptors) of the dendrimers. Furthermore, their unique architecture offers the possibility of being a smart delivery system for both small organic ligands and nanoscopic reagents such as DNA or antibodies. The general structure of dendrimers is shown in Figure 13.

In 2015, Zhong et al. followed this strategy, tumor-specific dendrimers, to ensure the specific delivery of HDACi to the cancer cells [55]. Compound **25** (Figure 14) consisted of SAHA as a HDACi and folate units, which were linked to a generation 5 (G5) polyamidoamine dendrimer. The ratio of 3.2 and 3.5 per each G5 dendrimer, for *SAHA* and folate units, respectively, was determined using NMR spectroscopy. In vitro evaluation using KB cells that overexpress folate receptor, proved significantly higher uptake of compound **25** in comparison to its folate-free analogue **26** (Figure 14), which was used as a control. The uptake of compound **25** was found to be blocked by the pretreatment with 100 µM folic acid, confirming its FR-specificity. Importantly, compound **25** demonstrated higher cytotoxic properties against KB cells than the analogue **26**. Furthermore, in contrast to free SAHA, compound **25** did not demonstrate any cytotoxic properties against the macrophage cell model RAW264.7, and therefore, it did not affect the immune system, increasing its therapeutic efficacy on cancer cells.

### 3.3. Nanoparticles-Based Tumor-Targeted HDAC Inhibitors

Nanoparticles (NPs) are other systems that improve the bioavailability and release of biologically active compounds. This strategy leads to the development of potential drugs with better clinical benefits. For example, in the case of cancer, targeted polymeric NPs can be used to deliver anticancer agents to tumor cells with increased efficacy and reduced cytotoxicity in normal tissues. It should be noted that this type of transporters can be formulated with biocompatible and biodegradable copolymers, making them excellent carriers for drugs. Beyond the anticancer agents, NPs can be applied to deliver a spectrum of diagnostic and imaging agents for several applications.

#### 3.3.1. Lipid-Based NPs

Specific delivery of HDACis into the tumor cells using NPs was achieved in 2009 by Ishii et al. [56]. For this purpose, five prodrugs, **28**–**32** (Figure 15), of the potent HDACi **27** (Figure 15) were used for the preparation of cationic NPs as a DNA vector to transfect plasmid DNA into human cells (human prostate cancer cells PC-3 and breast cancer cells Sk-Br-3). The obtained NPs were made up of cholesteryl-3β-carboxyamidoethylene-*N*-hydroxyethylamine (**33**), Tween 80, and the corresponding prodrug of **27** in a molar ratio of 85:5:10. It was noticed that the NPs containing **27**
*n*-dodecanoic acid derivatives demonstrated two to four times higher gene expression than NPs without **27** prodrugs in their structures. The authors indicated that the achieved enhancement of the gene expression may occur due to the hyperacetylation of histones caused by intact HDACi released from the prodrug into the cell-incorporated vector. 

In 2014, Foglietta et al. used *butyric*
*acid* as HDACi for the preparation of cholesteryl butyrate solid lipid NPs [57]. The authors indicated that the proposed delivery system can overcome the unfavorable pharmacokinetic and pharmacodynamic properties of *butyric acid*. During these studies, it was noticed that cholesteryl butyrate solid lipid NPs induced a higher and prolonged expression level of the butyrate target genes at lower concentrations than sodium butyrate. Additionally, these NPs led to a significant decrease in cell proliferation, as well as significant inhibition of total HDAC activity and overexpression of the p21 protein in HL-60 promyelocytic leukemia cancer cells. Surprisingly, the experiment with breast cancer cell line MCF-7 showed that cholesteryl butyrate solid lipid NPs did not improve the anticancer activity in comparison with free sodium butyrate, which was caused by the differences in the expression of the butyrate transporter SLC5A8 in the evaluated cell lines.

In 2020, Han et al. used *valeric acid* as HDACi for the preparation of lipid NPs [58]. *Valeric acid* has a wide spectrum of applications in medicine, for example in the treatment of insomnia and seizures [59]. During these studies [58] *valeric acid* also demonstrated growth inhibitory activity against numerous cancer cell lines, including liver cancer cell lines, e.g., Hep3B, in the millimolar range. Furthermore, encapsulation of *valeric acid* in cationic lipid NPs enhanced anticancer activity against liver cancer cells. For example, in an experiment with Hep3B cells, the inhibition rate after 72 h of the treatment with encapsulated *valeric acid* (67.82%) was more than 16% greater than that of free *valeric acid* (51.75%) at a concentration of 2 nM. The HDAC inhibitory properties of free and encapsulated *valeric acid* were tested in the HDAC activity assay. Interestingly, encapsulated *valeric acid* inhibited HDAC activity significantly more efficiently than free *valeric acid* in Hep3B cells after 72 h, with relative HDAC activity values of 0.168 and 0.315, respectively. Moreover, encapsulated *valeric acid* presented favorable anticancer properties in mouse models implanted by two liver cancer cell lines (Hep3B^Luc^ and SNU-449^Luc^).

#### 3.3.2. Polymer-Based NPs

Poly-lactide-co-glycolide acid (PLGA) was used to prepare PLGA-based NPs loaded with *SAHA* **34** (Figure 16) by Sankar R. and Ravikumar, V. in 2014 [60]. These NPs demonstrated good hemocompatibility. They did not elevate any of the biochemical parameters of the blood and did not present noticeable changes in the tissues of the examined organs (liver, kidney, lung, and heart) compared to the control. Spectrofluorometry and fluorescence microscopy techniques showed that the obtained NPs were detectable in the liver, kidney, and heart after 3 days. Furthermore, the obtained PLGA-based NPs bearing SAHA were shown to be actively absorbed into A549 lung cancer cells, indicating that such NPs can potentially be used in cancer treatment.

*SAHA* and *Quisinostat* **35** (Figure 17) were used by Wang et al. in 2015 for the synthesis of PLGA-lecithin-PEG core-shell NPs using PLGA, soy lecithin, and 1,2-distearoyl-sn-glycero-3-phosphoethanolamine-*N*-carboxy(polyethylene glycol) 2000. For the preparation of these NPs, a modified nanoprecipitation technique was used [61]. Two forms of PLGA polymers, ester- and carboxyl-terminated PLGA, as well as different percentages of lactide and glycolide moieties, were examined. Both *SAHA*- and *Quisinostat*-containing NPs released their active compounds within 4 days, with drug release rates of ~95%. Five cancer cell lines (HCT116, SW620, SW837, PC3, and DU145) were used to evaluate the radiosensitization properties of the obtained HDACi-NPs. NPs loaded with both HDACis were shown to increase the sensitivity of various solid tumor cell lines to radiotherapy. For example, the sensitizer enhancement ratio (SER) determined in HCT116 cells for *SAHA*-containing NPs was 1.71, while SER determined for free *SAHA* was 1.48. Interestingly, the radiosensitization properties of HDACi-containing NPs were confirmed in vivo. In an experiment using mice bearing PC3 flank xenograft tumors, which were treated with saline, free *SAHA*, and *SAHA*-containing NPs, followed by a single dose of radiotherapy, it was observed that *SAHA*-containing NPs had a therapeutic efficacy significantly higher than the *SAHA* free. Analogous results were achieved when *Quisinostat*-containing NPs were evaluated in an experiment using mice bearing subcutaneous flank xenografts of SW620. *Quisinostat*-containing NPs were significantly more efficient than the free *Quisinostat*.

*SAHA*-containing NPs were also identified as antiviral agents. In 2015, Tang et al. prepared PLGA-PEG NPs coated with Tm-cell specific scFv CD_45_RO antibody and loaded with *SAHA* and human immunodeficiency virus 1 (HIV-1) protease inhibitor *Nelfinavir* **36** (*Nel*) (Figure 17) [62]. CD4^+^ T-cells are known to be the main targets of the HIV-1, and thus, eradicating the latent form of the virus from such cells is the crucial issue in the treatment of AIDS [63]. Tang et al. [62] demonstrated that *SAHA*- as well as *SAHA* and *Nel*-containing NPs were able to target latently infected CD4^+^ T-cells and showed low in vitro toxicity against ACH-2 cells. Importantly, *SAHA* and *Nel*-containing NPs were able to simultaneously activate latent virus and inhibit viral spread, indicating the potential of the obtained agents to target and eliminate the latent source of HIV.

In 2016, el Bahhaj et al. showed that the use of NPs can lead to an improved and selective delivery of HDACis to cancer cells and to an enhancement of their anticancer properties in vivo. The authors used a Ring-Opening Metathesis Polymerization (ROMP) of azido-polyethylene oxide-norbornene macromers functionalized by a click reaction to obtain NP **37a**, which contains a *Trichostatin A* analogue **37b** (Figure 18) [64]. *Trichostatin A* is known to have HDAC inhibitory properties; however, due to its unfavorable ADME and toxicity properties, it is not used in clinic [65]. The protecting role of this delivery strategy was proved by a bioluminescence resonance energy transfer (BRET) assay in MeT-5A cells. Even though the HDAC inhibitory properties of compound **37a** were lower in comparison with free HDACi, the activity of compound **37a** was maintained for more than 48 h, whereas free HDACi demonstrated loss of 50% activity after 48 h of incubation. Using the orthotopic model of peritoneal invasive cancer, **37a** was shown to selectively accumulate into tumor cells. Furthermore, a combination of **37a** with *Decitabine* led to an 80% reduction of tumor weight in vivo, without exhibiting any toxic effects. Analogous therapy using free HDACi was found to be ineffective [64]. The same research group applied ROMP technique to develop other NPs containing HDACis, such as **14** (Figure 8), and norbornene-polyethylene oxide macromers [66].

In 2017, Zhang and coworkers developed novel functionalized NPs, such as *FA17-PLGA NPs* **38** containing *FA17-PLGA*
**39** (Figure 19). The goal of this strategy was to use these NPs for targeted delivery of cytotoxic agents based on the high HDAC concentrations in tumors. The modified PLGA **39** was obtained by conjugation of the PLGA and *FA17* as a HDACi [67]. Cellular uptake studies using a fluorescence assay and MCF-7 cells indicated that **38** could be internalized into MCF-7 in a time-dependent manner. This suggested that the conjugation of PLGA to *FA17* was beneficial for the internalization and active targeting ability of NPs. In vivo experiments with tumor-bearing mice showed selective and favorable in vivo distribution of NPs containing *FA17*. Reduced retention in the liver, a lower drug distribution in the liver, lung, and spleen, as well as a higher concentration in tumor tissues, were observed. Furthermore, the in vivo anticancer activity of compound **38** in combination with *Paclitaxel* (PTX) was examined in mice injected with MCF-7 cells. Interestingly, the treatment with *PTX-FPLGA NPs* exhibited the highest antitumor effect, which was remarkably greater than for PTX solution and combination of the PTX and NPs without *FA17* (*PTX-PLGA NPs*). The tumor inhibition rates after 13 days of treatment were 90%, 84%, and 65% for the *PTX-FPLGA NPs*, *PTX-PLGA NPs*, and PTX solution groups, respectively.

In 2017, Thapa et al. used a water-insoluble zein plant protein polymer to prepare *SAHA* and *Bortezomib* **40** (proteasomal inhibitor, Figure 20) combination-loaded zein NPs for the treatment of metastatic prostate cancers [68]. The obtained NPs were characterized by pH-dependent drug release profiles and exhibited high uptake in different prostate cancer cells. Importantly, treatment with zein NPs containing *SAHA* and *Bortezomib* demonstrated synergistic enhancement in anticancer efficacy in all prostate cancer cell lines examined (PC3, DU145, and LNCaP). Furthermore, an improved anti-migration effect and the ability to induce the pro-apoptotic proteins in the cells were also observed. Their anticancer properties were tested in a PC3 tumor xenograft mouse model, showing minimal toxicity and an antitumor effect greater than that observed for each free active agent.

Recently, in 2020, the development of PLGDA-based NPs containing *SAHA* and catalase (a biological peroxidase), using a double emulsion method, allowed to overcome the radiation resistance due to tumor microenvironments, including hypoxia and histone deacetylase (HDAC) overexpression [69]. These NPs protected the catalytic activity of catalase and prolonged the exposure to the HDACi. It was observed that NPs containing *SAHA* and catalase could reduce the intratumoral H_2_O_2_ concentration compared to the NPs loaded only with *SAHA*. After the injection of 25 µL of *SAHA*-containing NPs, the intratumoral oxygen concentration remained stable (almost no oxygen generation within the tumor tissue was observed), whereas, after injection of 25 µL of *SAHA*- and catalase-containing NPs, the oxygen concentration was raised to 55 µM in 25 min. Interestingly, the authors confirmed that such a synergistic strategy could sensitize radiation therapy to tumor cells by the production of oxygen through the biological activity of the catalase, and by the deregulation of HDACs through the application of the HDACi. An in vivo experiment that included the application of radiotherapy indicated that *SAHA*- and catalase-containing NPs led to the most effective tumor inhibition (87.28%). Furthermore, in vivo studies demonstrated the absence of toxic effects and the excellent biocompatibilities of the obtained NPs.

#### 3.3.3. Sugar-Based NPs

In 2019, Lee et al. conjugated the HDACi 4-phenylbutyric acid (PBA) **41** with the backbone of hyaluronic acid (HA) through an esterase cleavable moiety (Figure 21) [70]. The obtained PBA-HA NPs **42** were further loaded with *Curcumin* (CUR) to enhance the anticancer potential of CUR for lung cancer therapy. NPs of PBA-HA loaded with CUR showed a lower IC_50_ value compared to CUR or PBA-HA NPs in combination with free CUR in an in vitro assay with A549 cells (the IC_50_ values were 13.0, 18.2, and 15.9 µg mL^−1^, respectively). Furthermore, in vivo experiments indicated a higher accumulation of these NPs in tumor tissue and a lower distribution in liver and spleen in a A549 tumor-bearing mouse model. Multiple dosing of CUR-loaded PBA-HA NPs showed effective tumor growth suppression and apoptosis-inducing effects without significant difference in body weight and any histological changes in the normal tissues of mice compared to the control group.

In 2019, Alp et al. prepared NPs containing HDACi based on biocompatible starch [71]. *CG-1521* **44** (Figure 21) was selected as HDACi to be encapsulated in a starch matrix using the emulsion-solvent diffusion technique. After optimization of the physicochemical parameters of starch NPs loaded with *CG-1521* (size, zeta potential, morphology, loading, and release kinetics), cytotoxicity was tested. Importantly, the encapsulated *CG-1521* demonstrated a substantially reduced drug release rate and significantly enhanced cytotoxic capacity compared to free HDACi in MCF-7 cells. *CG-1521*-loaded starch NPs induced cell cycle arrest and significant apoptosis.

*Valproic acid* **45** [72] was also used as HDACi for the preparation of NPs by a nanoprecipitation and emulsification technique (Figure 22) [73]. In 2020, Lindemann et al. synthesized a series of polysaccharide NPs containing *valproic acid* based on a matrix of cellulose and dextran, which differed in the ratio of the loaded *valproic acid* units. The studies carried out indicated that the uptake of the selected *valproic c acid*-containing polysaccharide NPs occurred within a few seconds, locating NPs mainly in the cytoplasmic compartment of HeLa cells. Furthermore, these selected NPs were shown to be non-toxic to HEK-293T cells and led to a significant reduction of HDAC2 activity in the presence of lipase, which is necessary to catalyze the cleavage of the ester bond that results in the formation of free *valproic acid* from NPs **46**. More recently in 2021, the same research group synthesized NPs **47** (Figure 22) containing *valproic acid* also based on cellulose and dextran but modified with sulfate residues to improve intracellular drug release [74]. The newly synthesized NPs **47** were non-toxic in vitro and in vivo and demonstrated rapid cellular uptake. In addition, they were able to induce histone H3 hyperacetylation thanks to the inhibitory activity of HDAC.

In 2021, Chaudhuri et al. synthesized novel cyclodextrin (CD)-based nanoparticles containing β-amino ester and PEG units for the encapsulation of HDACis [75]. One example is compound **48,** a PEGylated CD nanoparticle (PEG-CDN) loaded with *Panobinostat* (Figure 23). An in vitro experiment of **48** with a murine glioblastoma GL261 cell line demonstrated an IC_50_ value of 0.56 µM, while the IC_50_ value for free *Panobinostat* was 0.17 µM. An in vivo study with mice injected with GL261 cells indicated that the treatment of tumors with PEG-CDN loaded with 30 µg of *Panobinostat* resulted in a significant reduction in tumor growth compared to PEG-CDN and saline controls. On the other hand, the median survival for *Panobinostat*-loaded PEG-CDN was 22 days, which was comparable to 20 days for PEG-CDN and saline controls, and therefore, the treatments did not significantly prolong survival of the animals.

## 4. Imaging Probes

Many studies have demonstrated that one of the main causes for deaths or poor prognosis in a disease is late diagnosis, showing its association with high morbidity and low survival [76,77,78]. Thus, research on more efficient treatments and selective methods that allow an early diagnosis are of key importance. Along this line, molecular imaging plays an important role in noninvasive earlier diagnosis, the accurate detection of diseases or dysfunctions, treatment follow-ups, personalized treatments, and it is also useful in drug development and discovery processes [79]. Among these techniques, positron emission tomography (PET) and fluorescence are the most widely used.

### 4.1. PET Ligands

As a non-invasive imaging technique, PET can be used for detection of various pathological changes in the living brain that arise due to neurodegenerative diseases (NDs), such as Alzheimer, Parkinson diseases, and cancer [76,80]. Importantly, epigenetic modifications, such as acetylation, methylation, ubiquitination, and phosphorylation, are crucial for proper central nervous system (CNS) function and, therefore, they can be monitored to detect NDs [81]. Since the acetylation of histones is regulated by the action of HATs and HDACs, HDACis appear as ideal ligands for the development of PET probes. All this supports the increased attention that the development of PET probes has received in the last decades.

Since 2006 many PET imaging agents for HDAC have been developed. Some examples (compounds **49**–**70**) are shown in Figure 24 [82,83,84,85,86,87,88,89,90,91,92,93,94,95,96,97]. They include radiolabeling with radioisotopes, such as ^11^C, ^18^F, and metal derivatives (^64^Cu), and most of them have been described in previous review articles and books [81,98,99] covering examples up to 2017. In this article, we focus on novel PET ligands of HDACs described from 2018 to the present.

In 2018, Kommidi et al. synthesized derivatives of *Panobinostat* radiolabeled with ^18^F [100]. The synthetic strategy followed to develop compounds **71** and **72** (Figure 25) was based on simple synthetic protocols that involved rapid, last-stage aqueous isotopic exchange ^18^F-radiochemistry. The biological activity of both compounds, in non-radiolabeled forms, was evaluated using diffuse intrinsic pontine glioma (DIGP) DIPG-IV, DIPG-XIII, and U87 (glistoblastoma) cells. Compound **71** effectively inhibited the growth of DIPG-IV, DIPG-XIII, and U87 cells in a nanomolar range (IC_50_ values of 122, 108, and 212 nM, respectively) and was only slightly less active than *Panobinostat* (IC_50_ values of 64, 38, and 65 nM, respectively). However, compound **72** was effective only against DIPG-XIII cells (IC_50_ value of 212 nM) and demonstrated significantly less efficacy against DIPG-IV and U87 cells (IC_50_ values of 4.358 and 4.483 µM, respectively). Both non-radiolabeled derivatives developed less toxicity to healthy astrocyte controls (IC_50_ values greater than 5 µM in both cases). Moreover, stability of compound **71** was assessed in an aqueous solution at physiological pH, and its distribution was followed by PET studies in vivo. Recently, compound **71** was tested in combination with the convection enhanced delivery (CED) strategy, where a cannula was implanted into the tumor by Tosi et al. [101]. These studies resulted in prolonged survival in the DIGP mouse model and showed the need for CED to achieve high brain concentration.

In 2018, Kim et al. used the synthetic click labeling approach for the synthesis of [^18^F]*FETSAHA* **73** (Figure 25) as an HDAC-targeted PET probe [102]. Studies with **73** showed high accumulation of the radioactivity in tumor tissue, rapid blood clearance and gastrointestinal track and renal excretion. Tumor-to-blood and tumor-to-muscle uptake ratios in the RR1022 sarcoma-bearing rat model were 1.21 and 1.83 at 30 min and 2.75 and 2.76 at 60 min, respectively. Furthermore, specific accumulation of **73** in the receptor was proved in an inhibition assay in the presence of an excessive amount of SAHA. The favorable tumor-imaging properties of **73** indicated that this compound can be used as an ideal PET probe.

In 2020, Li et al. developed PET imaging probes for tracking neurodegenerative and tumor diseases employing biphenyl benzamides as HDACs ligands [103,104,105]. One of the PET probes for HDACis that this group reported as CNS imaging agent was [^18^F]*INER-1577-3* **74** (Figure 25) [105]. Non-radiolabeled **74** demonstrated high inhibitory activity against HDAC6 (the K_i_ value of 0.005 µM) and HDAC8 (K_i_ value of 0.0097 µM) and was significantly less active against HDAC2 (K_i_ value of 0.166 µM). In vivo PET studies in rodents showed that the **74**-uptake peaked approximately 15 min after injection for the whole brain. It was proved that **74** can cross the blood–brain barrier (BBB) and can be used to monitor HDAC activities in vivo.

In 2020, Tago et al. used an HDAC6-selective inhibitor, *Tubastatin A* **75** (Figure 26), as the target-binding component for the synthesis of a ^18^F-labeled PET agent [106]. An inhibition assay with non-radiolabeled **76** in HDAC1 and HDAC6 gave IC_50_ values of 996 and 33.1 nM, respectively, demonstrating selectivity for HDAC6. Unfortunately, an in vivo biodistribution experiment showed low brain uptake of compound **76** in mice, which is a limitation for NDs. However, bone radioactivity was stable at around 2% ID/g after injection, and 70% of compound **76** was detected in plasma in unchanged form after 30 min. Both facts indicate high tolerance to defluorination.

The same research group, in 2021, developed ^18^F-labeled tetrahydroquinoline derivative **77** (Figure 26) as a brain HDAC6 imaging probe through copper-mediated radiofluorination from an arylboronic precursor [107]. Probe **77** demonstrated good penetration and moderate stability in the mouse brain, being rapidly metabolized to the corresponding carboxylic acid. The levels of radioactivity in blood, brain, liver, and kidney reached peaks of 4.51% 10 min post injection (p.i.), 7.86% 10 min p.i., 26.8% 30 min p.i., and 24.6% 60 min p.i., respectively. The brain/blood radioactivity ratio remained stable (~1.7) for 60 min. Inhibition assays with recombinant HDAC1 and HDAC6 enzymes indicated that non-radiolabeled compound **77** exhibited approximately 120-fold higher selectivity against HDAC6 than HDAC1 (IC_50_ value of 7 nM for HDAC6). Additional in vivo studies in the presence of selective HDAC6 inhibitors revealed displacement of more than 80% of compound **77** uptake in the brain, proving the selective binding of compound **77** to HDAC6.

In 2021, Turkman et al. developed a novel late-stage radiosynthesis for compounds containing a 5-trifluoromethyl-1,2,4-oxadiazole (TFMO) group, which is responsible for the specific inhibition of HDAC2 [108]. In this approach, late-stage incorporation of [^18^F]fluoride into the TFMO unit was achieved using a bromodifluoromethyl-1,2,4-oxadiazole moiety, which was transformed into [^18^F]TFMO through the exchange of no-carrier-added bromine-[^18^F]fluoride in a single step. Application of the developed radiosynthetic protocol led to PET ligands with good radiochemical yield (3–5%), high radiochemical purity (over 98%), and moderate molar activity (0.33–0.49 GBq/µmol). Furthermore, radiosynthesis of TFMO-containing potent HDAC2 inhibitor [^18^F]*TMP195* **78** (Figure 26) was performed as a validation process of the reported method. Despite the great importance of the approach presented as an innovative synthesis of radiofluorinated HDAC2 inhibitors, no data were communicated to demonstrate favorable biological and PET ligand properties of **78**.

The results obtained with PET ligands of HDAC in recent years highlights the need of developing new HDACi-PET candidates with increased biological activity and broader clinical potential. However, it is worth noting that some progress has been made in the biological evaluation of the discovered examples, as shown in the following examples. [^11^C]*Martinostat*
**62** (Figure 24) has been in vivo/in vitro cross-validated in pig brain for its ability to measure HDAC1-3 levels in vivo [109], and has entered in clinical trials with patients suffering from schizophrenia and schizoaffective disorders [110]. [^18^F]*TFAHA* **64** (Figure 24) has been used as non-invasive and quantitative imaging agent to detect class IIa HDACs expression-activity and pharmacological inhibition in rat intracerebral glioma models [111]. [^11^C]*KB631* **70** (Figure 24) has been evaluated as PET tracer for in vivo visualization of HDAC6 in B16.F10 melanoma in mice [112]. cGMP production and HDAC6 target occupancy measurement in non-human primates have been also performed using [^18^F]*Bavarostat* **68** (Figure 24) [113]. Furthermore, clinical validation of [^18^F]*Bavarostat* **68** has shown that it is safe and appropriate for quantifying HDAC6 expression in the human brain [114].

### 4.2. Fluorescent Probes

Due to the implication of HDACs in the epigenetic regulation of gene expression through the acetylation process, HDACis are of great importance as a starting point in the development of other imaging agents such as fluorescent probes. These agents can be a useful tool for obtaining preclinical images, before moving to other more expensive modalities thanks to its key advantages, such as low cost, ease of use, and multiplex imaging capabilities [115]. Application of such agents may lead to detect modifications of DNA and histones or enzymatic activity in living cells [116]. In recent years, great progress has been made in the development of fluorescent probes based on HDACi structures and many novel compounds have been reported.

In 2009, Mazitschek et al. synthesized a fluorescence-labeled, non-selective HDACi **79** (Figure 27) based on the structure of *SAHA* as the target-binding component. This strategy allows to measure HDACs binding by fluorescence polarization [117]. Compound **81** effectively bound HDAC3/NcoR2 and HDAC6 with K_d_ values of 2.0 and 4.6 nM, respectively. Compound **79** was used in a high-throughput screening (HTS) for the identification of HDACs inhibitors that target the active site, by correlation with data derived from assays showing enzyme turnover of a fluorogenic substrate. Thus, **79** was successfully used to assess the affinity of various known HDACi (e.g., *SAHA* and *Trichostatin A*) to HDACs in a HeLa nuclear extract, as well as in purified HDAC3/NcoR2 and HDAC6. In 2011, Singh et al. developed another *SAHA*-based fluorescent probe, *c-SAHA* **80** (Figure 27), to determine the binding affinities and dissociation rates of the enzyme-inhibitor complexes using the fluorescent displacement method [118]. This new derivative consisted of *SAHA* conjugated to coumarin and was applied to examine the affinity of several HDACis (e.g., *SAHA* and *Trichostatin A*) for HDAC8. The described method is applicable to most HDAC isozymes and can be easily adopted for HTS assays of HDACis.

Kong et al., in 2011, developed the fluorescent targeting probes containing a dansyl group and an HDACi as the imaging and the target-binding component respectively [119]. Among all the synthesized fluorescent probes, which differ in the length of the aliphatic chain, compound **81** (Figure 28) was selected as the most promising derivative. The inhibitory activities of **81** against HDAC1, -2, -3, -4, -5, -6, -7, -8, -10, and -11 (IC_50_ values of 0.95, 1.38, 1.12, 0.33, 0.40, 0.13, 2.56, 3.98, 0.42, and 0.48 µM, respectively) were similar to the activity of *SAHA* (IC_50_ values of 0.22, 0.56, 1.79, 0.64, 0.13, 0.027, 0.99, 2.74, 0.11, and 0.082 µM, respectively). These results indicate that compound **81** specifically targeted HDAC4 and HDAC6. Compound **81** inhibited the growth of three human prostate cancer cell lines, PC3, C4-2, and LNCaP, with IC_50_ values of 1.54, 1.91, and 1.30 µM, respectively. Furthermore, imaging of the PC3 and A549 cell lines using confocal microscopy indicated that **81** was accumulated in the cytoplasmic compartments of treated cells, but not in the nuclei. Interestingly, an increase in nuclear acetylation was also observed.

In 2015, Meng et al. used *Panobinostat* as the target-binding component for the development of new near-infrared fluorescence probes of HDACs. *Panobinostat* was coupled with *Cyanine-5.5* (Cy5.5) as a fluorescent dye to form probe *LBH589-Cy5.5* **82** (Figure 28) [120]. Compound **82** demonstrated high inhibitory activity against HDACs with an IC_50_ value of 9.6 nM when evaluated in an assay with HDACs-rich nuclear extract of HeLa cells. In vitro fluorescence microscopic studies indicated specific binding of **82** to HDACs in MDA-MB-231 cells. Additional imaging studies with MDA-MB-231 tumor xenografts mice showed an accumulation of **82** in the tumor with satisfactory contrast from 2 h to 48 h after injection. The fluorescent signal of the probe in tumor tissue was successfully reduced by co-injection with non-fluorescent *Panobinostat*. Moreover, the therapy evaluation study with *SAHA* demonstrated the potential application of the developed probe for the evaluation of the therapeutic response to HDACi in cancer therapy.

Shin et al. synthesized three fluorescent probes based on benzamide derivatives **83a**–**c** (Figure 28) with different polyaromatic substituents (naphthalene, anthracene, and pyrene) [121]. The obtained probes **83a**–**c** showed red absorption and high fluorescence efficiency (fluorescence quantum yields (Φ_f_) values in a range of 0.08074 to 0.315, when evaluated in methanol), which was dependent on the size of the aromatic group. Unfortunately, the synthesized derivatives showed poor inhibitory activity against HDAC1 (IC_50_ values in a range of 45 to 105 µM), indicating that the substitution of benzamide derivatives with polyaromatic groups is not beneficial for the HDAC inhibitory properties.

In 2015, Fleming et al. used *Scriptaid* **84** (Figure 29) as a precursor for the synthesis of the morpholine analog *4MS* **85** (Figure 29) [122]. In an inhibitory assay with a panel of HDAC isoforms 1, 3, 6, 8, and 11, compound **85** presented similar activity (IC_50_ values of 1.43, 0.32, 0.012, 1.81, and 0.29 µM, respectively) to *Scriptaid* (IC_50_ values of 1.74, 0.37, 0.012, 1.52, and 0.36 µM, respectively). Both *Scriptaid* and *4* **85** showed modest selectivity towards HDAC6, but **85** developed slightly higher growth inhibitory activity (IC_50_ = 0.29 µM) against acute myeloid leukemia KASUMI-1 cells than *Scriptaid* (IC_50_ = 0.49 µM). In addition, evaluation of **85** as a fluorescent probe was also performed. A rapid cellular uptake of **85**, which led to obtaining sufficient visualization via confocal microscopy, appeared in less than 50 s, when evaluated in MDA-MB-231 cells with a 1.0 µM concentration of **85**. Interestingly, no fluorescence was observed in the nucleus, indicating that **85** did not penetrate the nuclear envelope.

The same research group synthesized another series of HDACi probes based on the chemical structure of *Scriptaid* [123]. Compound **86** (Figure 29) was found to be the most promising agent, with an inhibitory activity against HDAC 1, 3, 6, 8, and 11 similar or higher (IC_50_ values of 1.98, 0.36, 0.0035, 2.46, and 0.15 µM, respectively) than the parent *Scriptaid* (IC_50_ values of 1.74, 0.37, 0.012, 1.52, and 0.36 µM, respectively). Furthermore, compound **86** was more than 500-fold selective for HDAC6 compared to HDAC1 and 100-fold more selective against HDAC3. Compound **86** also demonstrated growth inhibitory activity in KASUMI-1 cells, with an IC_50_ value of 0.36 µM. Photophysical evaluation of **86** indicated strong fluorescent properties, with Φ_f_ values of 0.81 and 0.38 in dimethyl sulfoxide (DMSO) and buffer, respectively (higher than Φ_f_ of **85**, which were 0.03 and <0.01 in DMSO and buffer, respectively). The high fluorescence properties of compound **86** was confirmed in a cellular assay with A549 cells, as well as in whole organism imaging in a zebrafish model.

The same research group synthesized a third generation of fluorescent probes, **87a**–**c** (Figure 29) based on *Scriptaid* [124]. These new agents were obtained by the direct Buchwald–Hartwig amidation in position 3 of the aromatic part of *Scriptaid* and demonstrated higher activity than *Scriptaid* against several HDACs isoforms. For example, all compounds were found to be very potent HDAC6 inhibitors with an IC_50_ values in the range of 0.58 to 1.0 nM (IC_50_ of *Scriptaid* was 12 nM). Although inhibitory activity of compounds **87a**–**c** against other HDACs was also significant, their high selectivity against HDAC6 is notable. For example, the selectivity factors of compounds **87a**–**c** ranged from 38 to 150 for HDAC6 over HDAC1. In addition, the whole cell tubulin deacetylation assay indicated that **87a**–**c** were more potent than *Tubastatin A* (Figure 26). Their optical properties (Φ_f_ values in a range of 0.03 to 0.05) made them suitable for cell imaging studies and theragnostic applications.

Based on the chemical structure of **85**, in 2017, Zhang et al. developed two derivatives as novel fluorescent probes with selective inhibition of HDAC6 [125]. During these studies, compound **88** (Figure 29) was found to be a selective HDAC6 inhibitor, inducing hyperacetylation of tubulin but not of histone H4. The IC_50_ values of compound **88** towards HDAC1-3 were greater than 20 µM, while the IC_50_ value against HDAC6 was 0.139 µM. Furthermore, fluorescent and immunofluorescent studies with A549 cells co-treated with proteasome inhibitor indicated that compound **88** was able to selectively target and image HDAC6 concentrated in aggresomes.

In 2017, Meyners and coworkers developed a series of novel fluorescence probes that combine the [1,3]dioxolo[4,5-*f*]benzodioxole core with a trifluoromethylketone residue as imaging and target-binding components, respectively [126]. To demonstrate their imaging capabilities, they used a lifetime fluorescence-based binding assay for HDACs inhibitors of class IIa. All synthesized compounds developed HDAC inhibitory properties against a pool of HDACs. In general, compound **89a** (Figure 30) showed the highest HDAC inhibitory properties with IC_50_ values of 0.82, 3.2, 0.042, 0.010, 0.10, 0.79, and 0.021 µM against HDAC1, -2, -3, -4, -5, -6, and -8, respectively. Compound **89b** (Figure 30) demonstrated the highest activity against HDAC7 (the IC_50_ value of 0.030 µM). These probes were further used to develop and perform a screening assay with several known HDACi (e.g., *SAHA* and *Trichostatin A*).

In 2018, Ho et al. reported a series of selective fluorescent HDAC6 inhibitors based on a naphthalimide core as the imaging component [127]. All the synthesized compounds demonstrated very potent HDAC6 inhibitory properties, with **90** (Figure 30) showing the highest activity and selectivity towards HDAC6. In a luminescent assay with HDAC1 and HDAC6, IC_50_ values towards HDAC1 and HDAC6 of **90** were 108 and 0.1 nM, respectively, with a HDAC6/1 selectivity index of 1080. This compound showed also considerable selectivity for HDAC6 over HDAC2 (1620-fold), HDAC3 (640-fold), and HDAC10 (3750-fold) and inactivity against HDAC8 and HDAC11. In vitro inhibitory activity of obtained compounds was tested in a pool of cancer cells, U87MG, MDA-MB-231, MDA-MB-468, MM.1s, LNCap, and PANC1. Overall, **90** showed the highest antiproliferative activity against most of the used cancer cell lines. Moreover, **90** was found to be more effective compared to *SAHA* against all cancer cell lines. Due to the presence of the 4-methoxy group in the structure, **90** presented the best photophysical properties (Φ_f_ value of 0.59) and was used for an imaging assay in MDA-MB-231 cells, showing that **90** was accumulated mainly in the cytoplasm and did not penetrate the nucleus.

In 2019, Radszus et al. developed fluorescent probes **91**–**93** (Figure 31), containing peptoids as the HDACi and dansyl group as the fluorescent tag [128]. Among them, compound **92** demonstrated the highest HDAC inhibitory activity against HDAC1, -2, -3, and -6 (IC_50_ values of 0.034, 0.081, 0.087, and 0.046 µM, respectively) and was also found to be more active than *SAHA* (IC_50_ values of 0.089, 0.183, 0.105, and 0.038 µM, respectively). Additionally, compound **92** had the highest antiproliferative activity against esophageal adenocarcinoma OE33 and OE19 cells (IC_50_ values of 0.776 and 0.925 µM, respectively). All probes showed favorable photophysical properties and their cellular uptake was monitored by fluorescence microscopy. Compound **91** showed the fastest uptake kinetic and the highest absolute fluorescence intensity in OE33 and OE19 cells at the concentration of 85 of 1.25 µM.

In 2020, Zhou et al. designed and synthesized a class of environmentally sensitive fluorescent HDACis, which can be a useful tool in the diagnosis and theragnosis of diseases associated with HDAC activity [129]. Four compounds, **94a**–**d** (Figure 32), which differ in the length and nature of the alkyl chain, were described. Assays with HeLa cell extracts indicated comparable inhibitory properties for compounds **94a** and **94b** (IC_50_ values of 205 and 109 nM, respectively) and *SAHA* (IC_50_ value of 135 nM), whereas compounds containing urea moiety were much less active or completely inactive, indicating that the presence of urea groups reduced the inhibitory activity. In an inhibitory assay against HDAC1, -2, -3, -6, -7, and -8, both **94a** and **94b** exhibited nanomolar activity and isoform selectivity comparable to *SAHA*. An in vitro assay with four cell lines, MOLT4, K562, PC-3, and A549, indicated that these compounds were generally less cytotoxic than *SAHA* and, therefore, beneficial for cell imaging. All compounds demonstrated good fluorescent properties but with high sensitivity to the environment. Thus, although their Φ_f_ values in a phosphate buffer saline solution were low, these values were increased by more than 100 times in an acetonitrile solution. Compound **94b** imaging assays showed fluorescence in cells with high expression of HDACs, PC-3, and MOLT4, and low fluorescence in cells with low expression of HDACs and HEK-293. Furthermore, compound **94b** was successfully used for the detection of mouse tumor tissue sections.

Tang et al. developed two *SAHA*-based fluorescence probes, **95** and **96** (Figure 33), where *SAHA* was conjugated to fluorescein isothiocyanate (FITC) or IRDye800CW as the imaging ligands, respectively. The purpose of these probes was to assess their potential in fluorescence image-guided surgery in hepatocellular carcinoma (HCC) [130]. Compound **95** was developed as an in vitro imaging agent to explore its targeting ability in HDAC-overexpressing cells. Compound **96** was synthesized as an agent for effective visualization of tumor lesions and to guide HCC resection during surgery. Initially, the authors used an HDAC Assay Kit to confirm the HDAC inhibitory properties of both compounds. The IC_50_ values for **95** and **96** were 0.219 and 0.193 µM, respectively (an IC_50_ value of 0.196 µM was measured for *SAHA*). Both compounds were completely non-toxic to HCC Bel-7402 and normal liver LO2 cells, indicating that they are safe as bio-probes for in vitro and in vivo imaging. Further in vitro studies demonstrated specific uptake of **95** by HDAC-overexpressed HCC Bel-7402 cells. Regarding **96**, in vivo studies proved its rapid accumulation with high tumor-to-background contrast on the subcutaneous and orthotopic HCC mouse model. Furthermore, orthotopic HCC was successfully resected using fluorescence image-guided surgery using **96**. The obtained results indicated that this compound is an optimal clinical translatable probe for the detection and resection of HCC.

Based on the chemical structure of *SAHA*, Huang et al. designed and synthesized a new fluorescent HDACi **97** (Figure 33) to be used as a probe for imaging tumor cells [131]. The HDAC inhibitory activity of **97** against HDAC1, HDAC3, and HDAC6 (IC_50_ values of 0.36, 0.57, and 0.19 µM, respectively) were comparable to *SAHA* (IC_50_ values of 0.19, 0.19, and 0.15 µM, respectively). Compound **97** was also found to selectively increase the acetylation of α-tubulin in a dose-dependent manner in MDA-MB-231 cells, while its effect in Hela cells was significantly lower. Compound **97** was successfully used to image MDA-MB-231, demonstrating its ability to efficiently penetrate cells. Fluorescence intensity was unchanged over 24 h.

In 2021, Yan et al. synthesized two new fluorescent probes, **99** and **100,** by conjugation of the highly selective HDAC8 inhibitor **98** to 1,8-napthalimide through the amide or aniline nitrogen, respectively (Figure 34) [132]. Both agents demonstrated good fluorescence properties and exhibited higher selectivity towards HDAC8 (K_D_ values of 8.05 × 10^−6^ and 7.42× 10^−6^ M, respectively) compared to **98** (K_D_ value of 6.25 × 10^−5^ M). Compound **100** had weak inhibitory effects on MDA-MB-231 and SH-SY5Y cells. Probe **99** was also inactive against MDA-MB-231 cells but demonstrated antiproliferative activity against SH-SY5Y cells with an IC_50_ value of 5.55 µM (IC_50_ value of **98** was 4.42 µM). Imaging assays also demonstrated the ability of **99** to target HDAC8 in MDA-MB-231 and SH-SY5Y cells and was successfully used to image SH-SY5Y tumor tissue sections.

## 5. Conclusions

HDACs are involved in chromatin remodeling and gene transcription mechanisms and their overexpression is associated to many pathological states and to the evolution of a wide range of diseases, including cancer and NDs. For this reason, HDAC has become an important target in the search of new treatments based on the development of potent HDAC inhibitors and degraders. On the other hand, rapid identification of HDACs expression using HDAC targeting probes is crucial for the early diagnosis and efficiency of many therapies.

This review reports the recent discoveries and innovative strategies used in the development of compounds that possess inhibitory or degradation activity against HDACs. We have paid special attention to the reported efforts on the synthesis of HDACs degrading compounds (PROTACs) and tumor-targeted HDACis (folate conjugates, dendrimers, and nanoparticles). In addition, we show some examples of novel HDACs imaging ligands, such as PET and fluorescent probes, which are useful and effective tools in early diagnosis.

To confirm their favorable therapeutic properties, many of the compounds described in this review require more detailed preclinical and clinical investigations. For example, in the case of PROTACs, their bioavailability is one of the challenges that must be addressed to achieve good candidates for in vivo experiments and clinical trials. However, recently developed ligands that target HDAC through innovative strategies have already shown a significant improvement in their therapeutic properties compared to conventional HDACis and may be of great importance for the treatment of many diseases. Regarding imaging agents, different *PET* and fluorescent probes with good imaging properties have been successfully obtained according to in vivo studies and have the possibility of being clinically translatable.

We hope that the examples collected in this review will inspire researchers in this area to investigate new and more effective strategies for the development of innovative drugs and imaging probes based on HDAC ligands.

## 6. Materials and Methods

The bibliographic search was carried out using the Google and Google Scholar search engines combined with different databases such as PubMed (https://pubmed.ncbi.nlm.nih.gov (accessed on 30 October 2021)), Web of Science (www.webofscience.com (accessed on 30 October 2021)), or Scopus (www.scopus.com (accessed on 30 October 2021)). All the chemical structures were drawn using ChemDraw 19 (www.perkinelmer.com (accessed on 28 December 2021)). Graphical abstract representation was created using BioRender (www.biorender.com (accessed on 28 December 2021)).

## Figures and Tables

**Figure 1 molecules-27-00715-f001:**
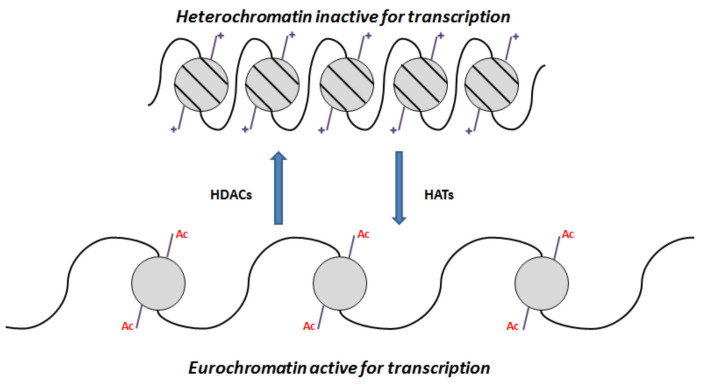
Chromatin regulation of transcriptional activity. Histone deacetylation induces the closure of the chromatin and acetylation induces an open chromatin structure.

**Figure 2 molecules-27-00715-f002:**
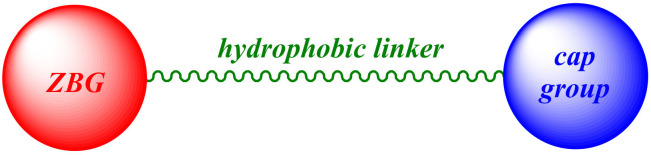
General structure of HDACis.

**Figure 3 molecules-27-00715-f003:**
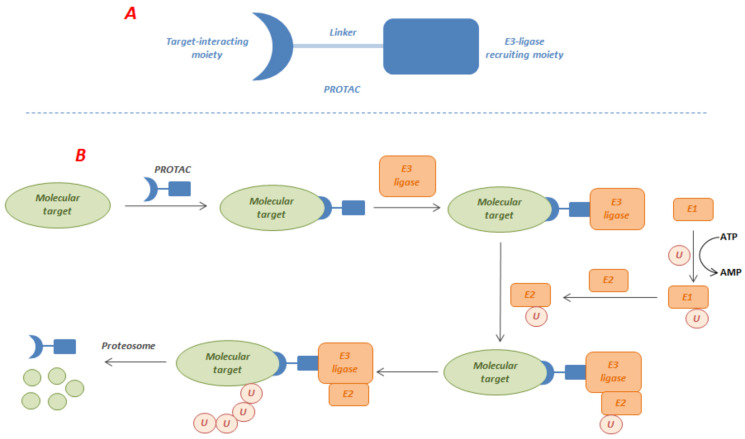
General structure of PROTACs (**A**) and their putative mechanism of action (**B**).

**Figure 4 molecules-27-00715-f004:**
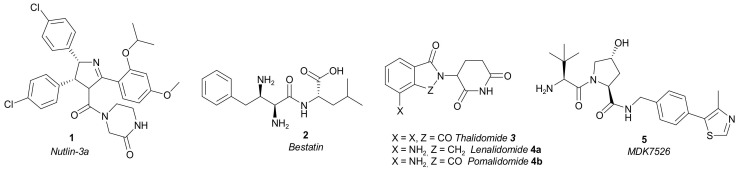
Examples of small molecules used as E3-ligase recruiters: *Nutlin-3a* **1**, *Bestatin* **2**, *Thalidomide* **3**, *Lenalidomide* **4a**, *Pomalidomide* **4b**, and *MDK7525* **5**.

**Figure 5 molecules-27-00715-f005:**
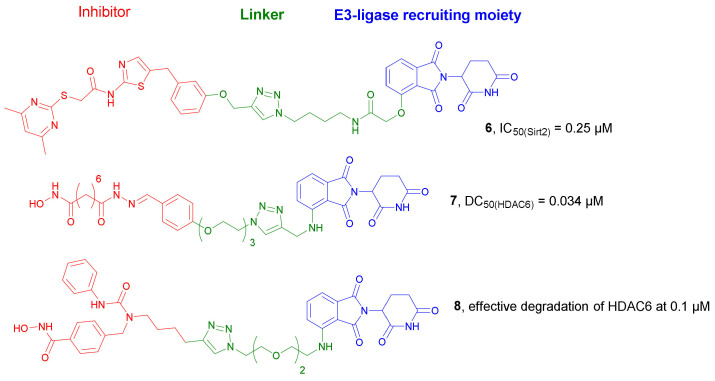
General structures of PROTACs **6**–**8**.

**Figure 6 molecules-27-00715-f006:**
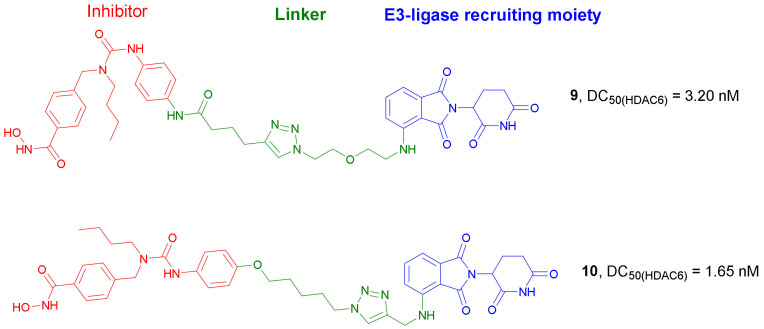
General structures of PROTACs **9** and **10**.

**Figure 7 molecules-27-00715-f007:**
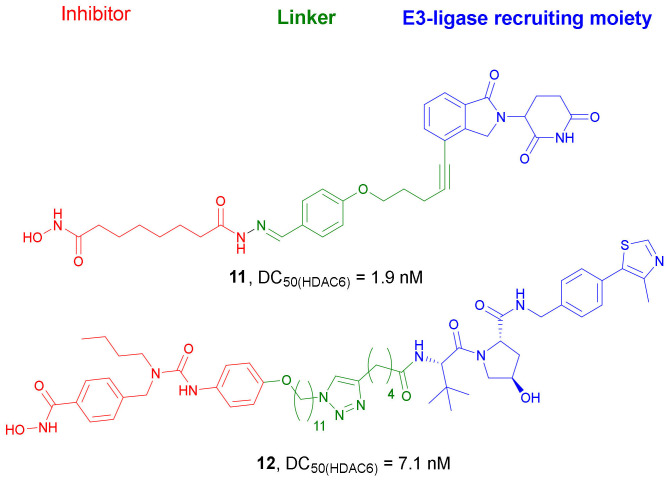
General structures of PROTACs **11** and **12**.

**Figure 8 molecules-27-00715-f008:**
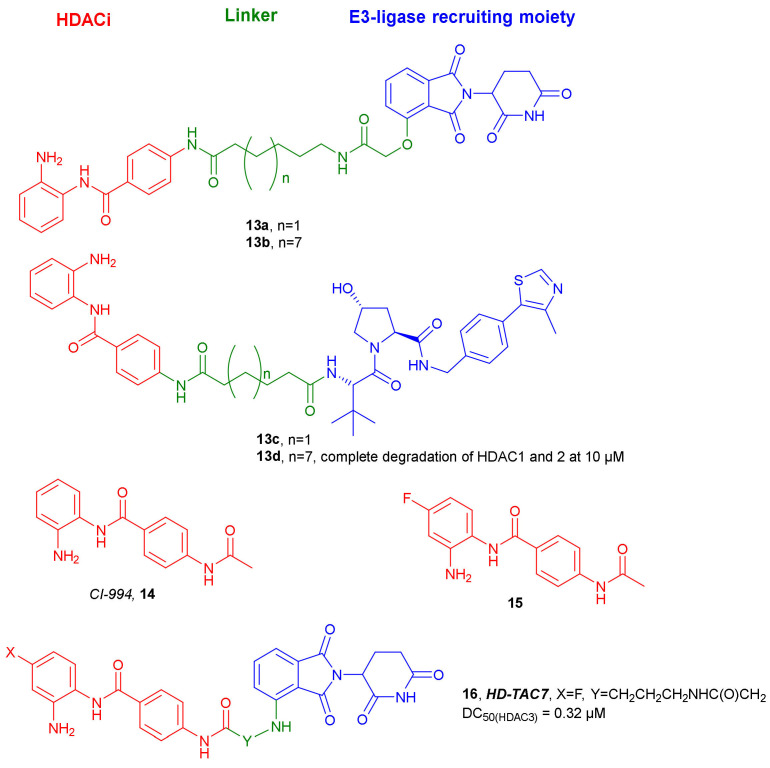
General structures of compounds **13**–**16**.

**Figure 9 molecules-27-00715-f009:**
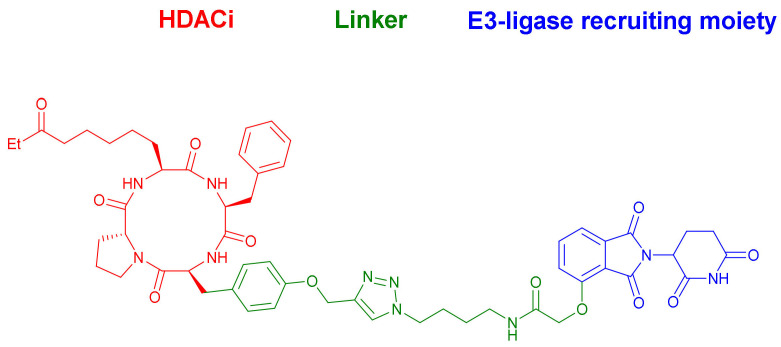
Chemical structure of PROTAC **17**.

**Figure 10 molecules-27-00715-f010:**
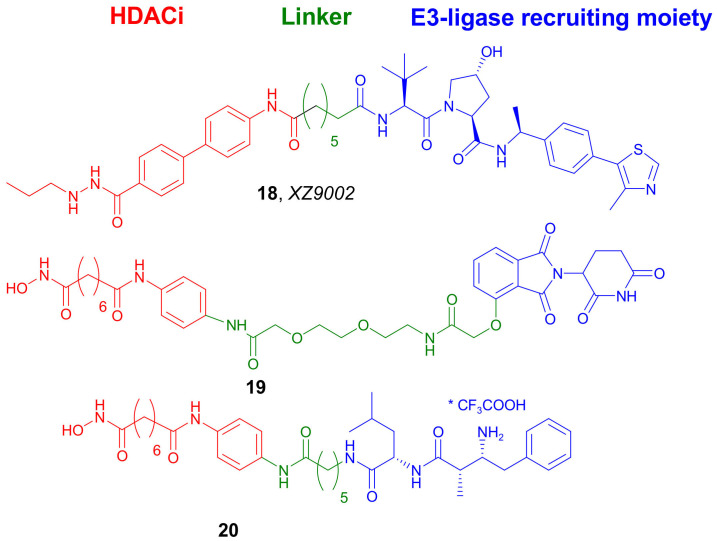
Chemical structure of PROTACs **18**–**20**.

**Figure 11 molecules-27-00715-f011:**
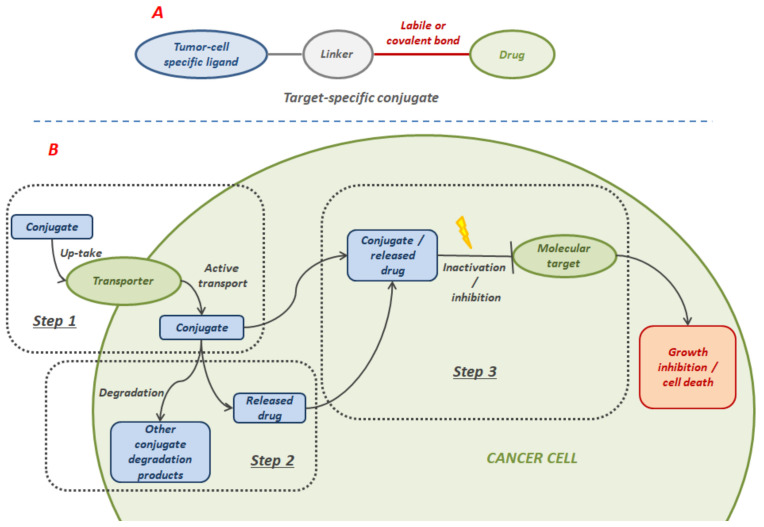
The general structure of tumor-specific conjugates (**A**) and their putative mechanism of uptake and action (**B**).

**Figure 12 molecules-27-00715-f012:**
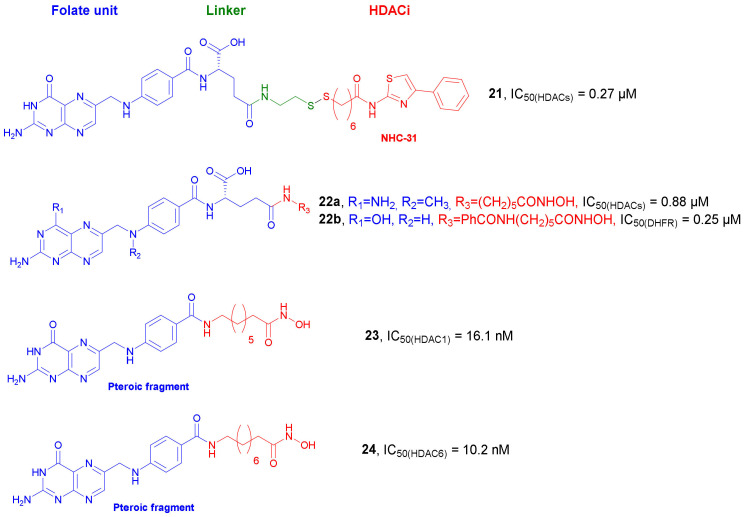
Chemical structure of HDACi -folic acid conjugates **21**–**24**.

**Figure 13 molecules-27-00715-f013:**
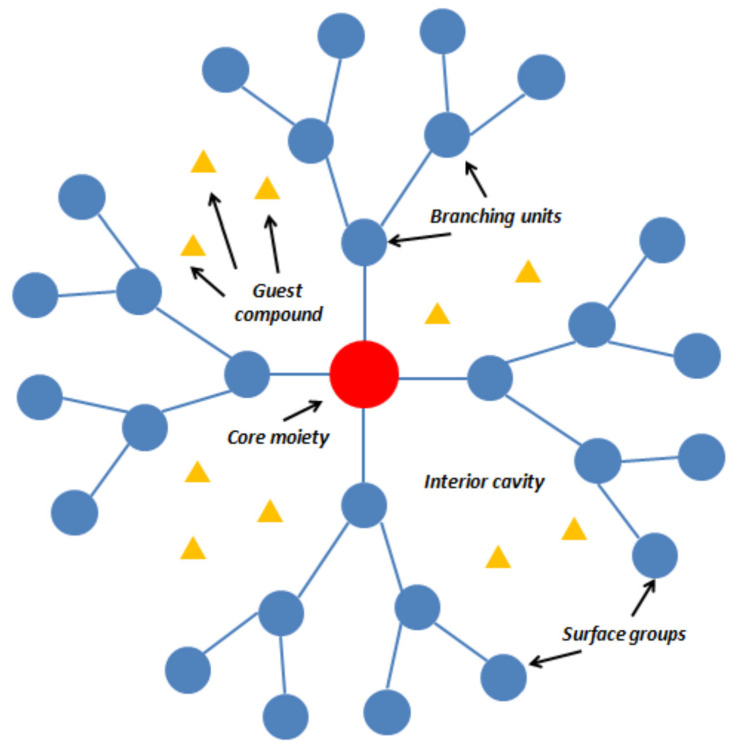
General structure of dendrimers.

**Figure 14 molecules-27-00715-f014:**
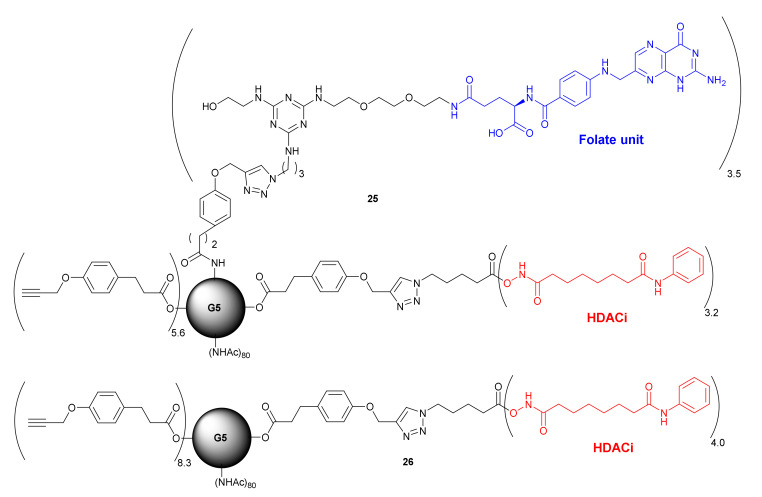
Chemical structure of HDACi-containing dendrimers **25** and **26**.

**Figure 15 molecules-27-00715-f015:**
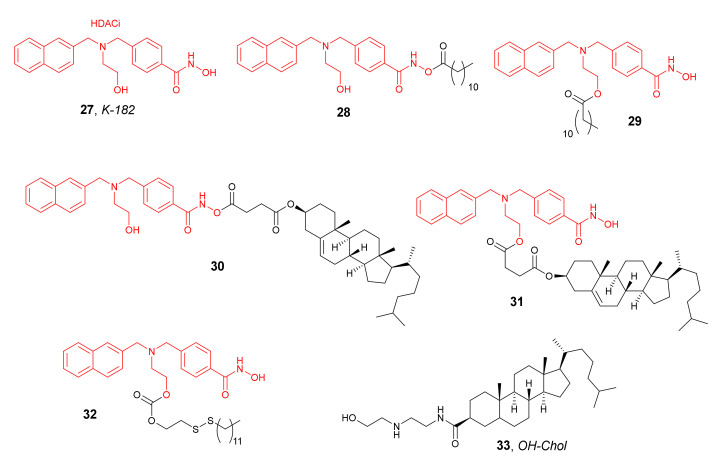
Chemical structures of **27**
*KI-182*, *KI-182* prodrugs **28**–**32**, and **33**
*OH-Chol*.

**Figure 16 molecules-27-00715-f016:**
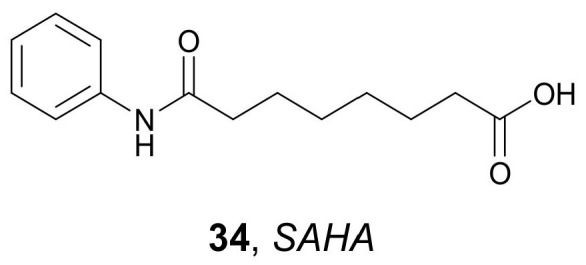
Chemical structure of *SAHA* **34**.

**Figure 17 molecules-27-00715-f017:**
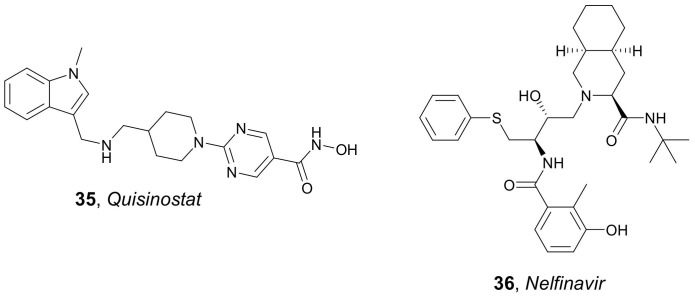
Chemical structures of *Quisinostat* **35** and *Nelfinavir* **36**.

**Figure 18 molecules-27-00715-f018:**
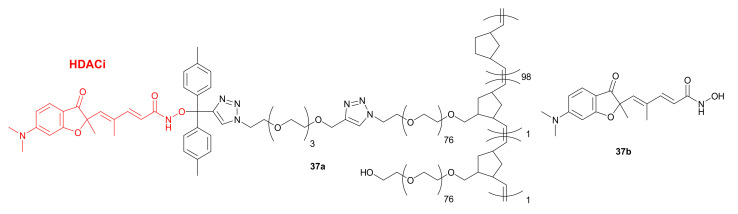
Chemical structures of compounds **37a** and **37b**.

**Figure 19 molecules-27-00715-f019:**
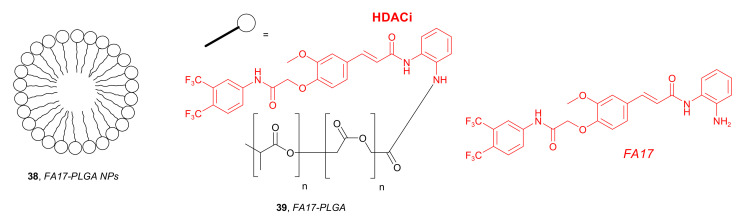
Chemical structures of *FA17-PLGA NPs* **38**, *FA17-PLGA* **39**, and *FA17*.

**Figure 20 molecules-27-00715-f020:**
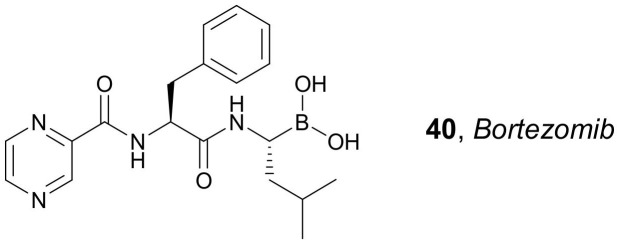
Chemical structure of *Bortezomib* **40**.

**Figure 21 molecules-27-00715-f021:**
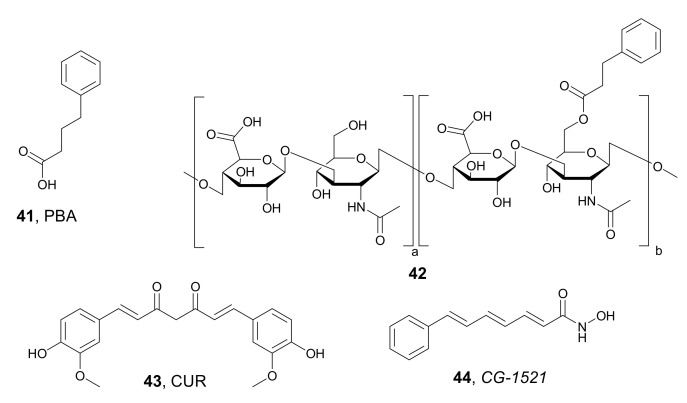
Chemical structures of PBA **41**, its HA-based NPs **42**, CUR **43**, and *CG-1521* **44**.

**Figure 22 molecules-27-00715-f022:**
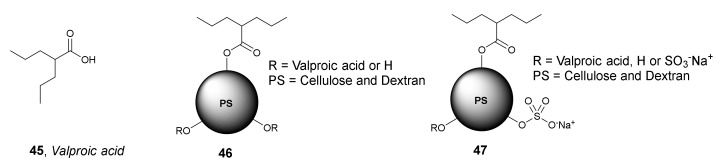
Chemical structures of *valproic acid* **45** and its polysaccharide-based NPs **46** and **47**.

**Figure 23 molecules-27-00715-f023:**
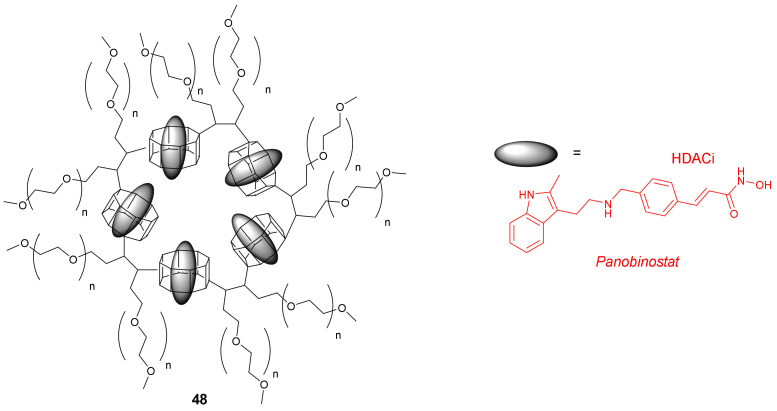
Chemical structures of CD-based and HDACi-containing nanoparticle **48** and *Panobinostat*.

**Figure 24 molecules-27-00715-f024:**
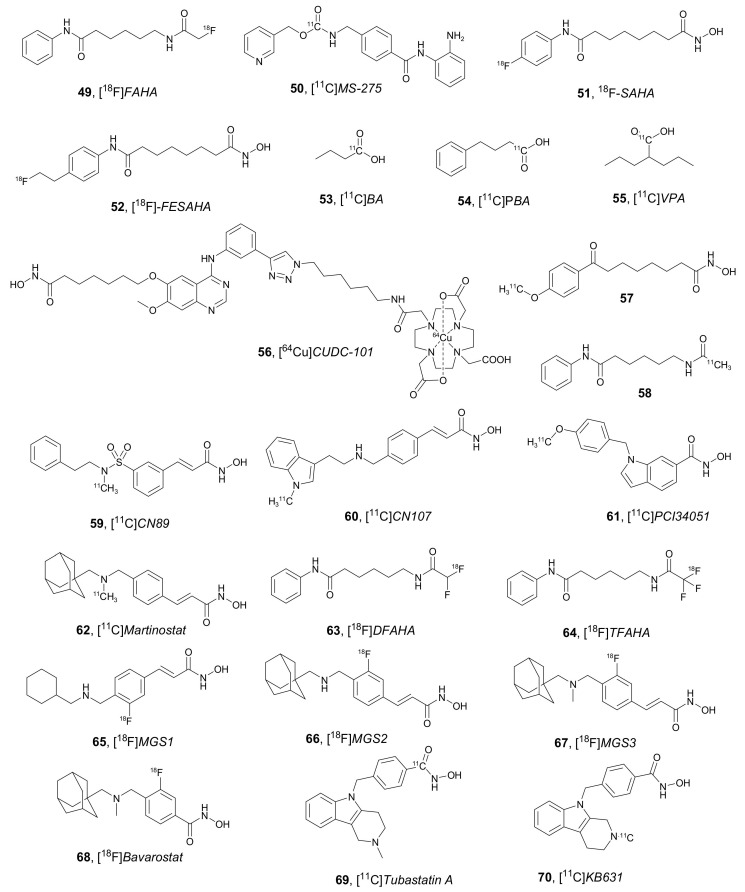
Chemical structures of PET ligands that possess HDAC affinity (compounds **49**–**70**).

**Figure 25 molecules-27-00715-f025:**
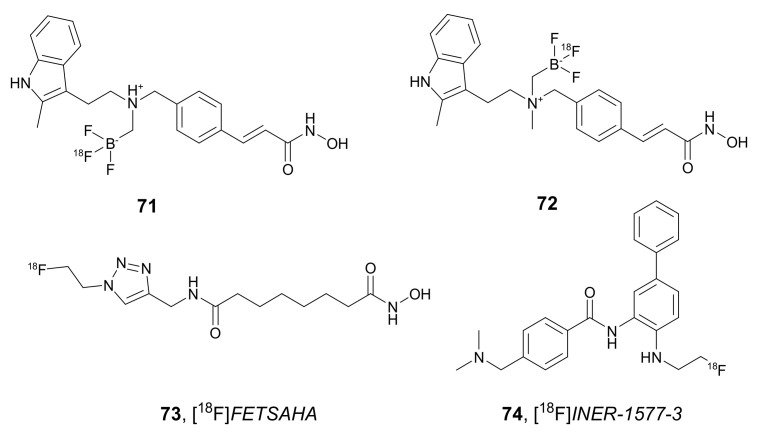
Chemical structures of compounds **71**–**74**.

**Figure 26 molecules-27-00715-f026:**
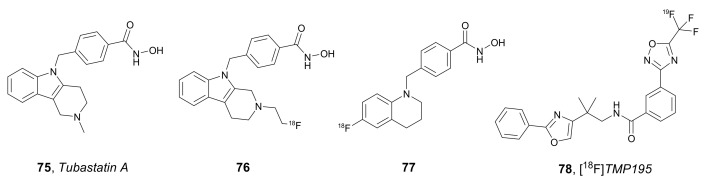
Chemical structures of compounds **75**–**78**.

**Figure 27 molecules-27-00715-f027:**
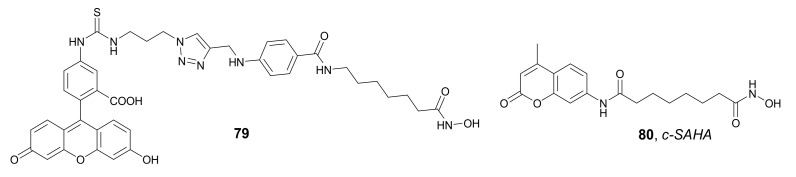
Chemical structures of compounds **79** and **80**.

**Figure 28 molecules-27-00715-f028:**
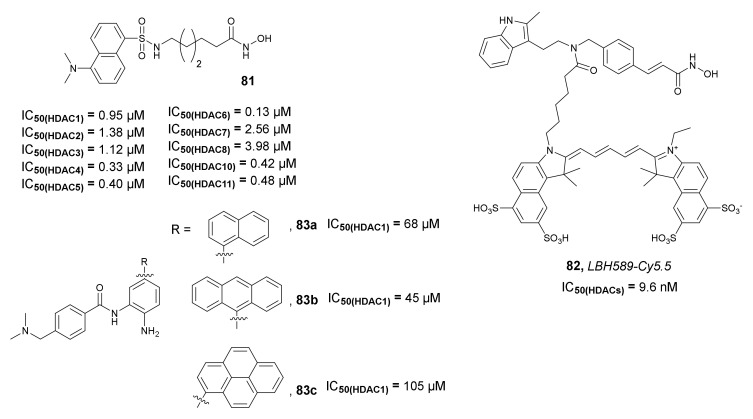
Chemical structure of compounds **81**–**83**.

**Figure 29 molecules-27-00715-f029:**
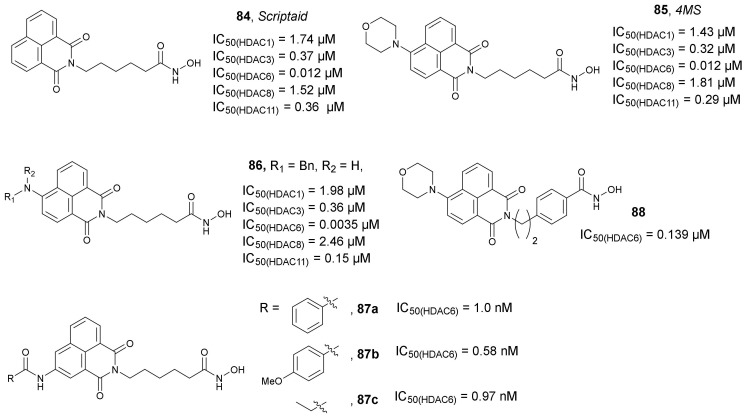
Chemical structures of compounds **84**–**88**.

**Figure 30 molecules-27-00715-f030:**
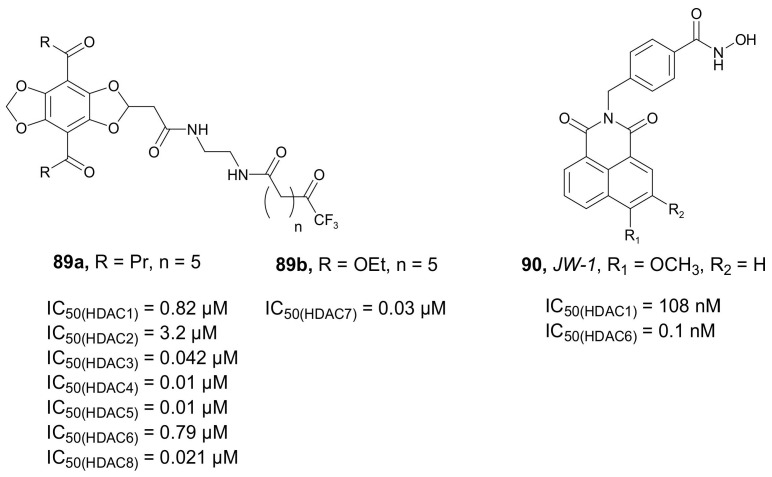
Chemical structures of compounds **89a**–**f** and **90**.

**Figure 31 molecules-27-00715-f031:**
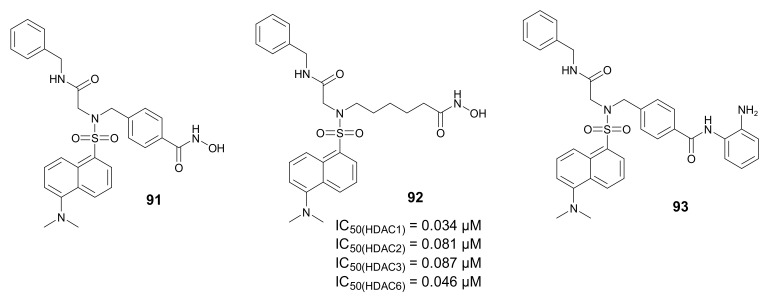
Chemical structures of compounds **91**–**93**.

**Figure 32 molecules-27-00715-f032:**
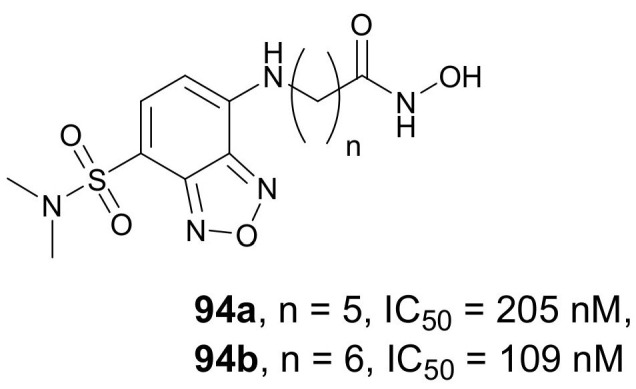
Chemical structures of compounds **94a** and **94b**.

**Figure 33 molecules-27-00715-f033:**
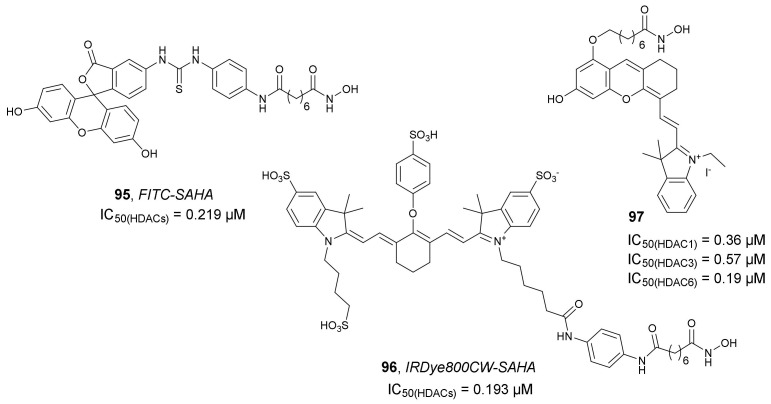
Chemical structures of *FITC-SAHA* **95**, *IRDye800CW-SAHA* **96**, and compound **97**.

**Figure 34 molecules-27-00715-f034:**
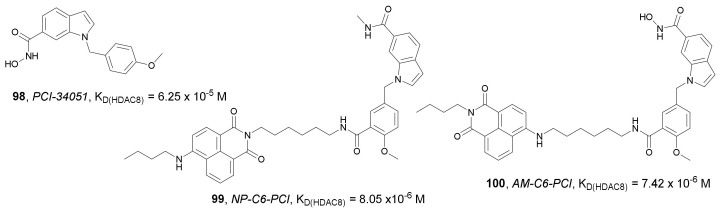
Chemical structures of *PCI-34051* **98**, *10NP-C6-PCI* **99**, and *13AM-C6-PCI* **100**.

## Data Availability

No Data.

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
