# Peer review of "HDAC Inhibitors: Innovative Strategies for Their Design and Applications"

_molecules, 2022, doi:10.3390/molecules27030715_

Round 1

Reviewer 1 Report

Targeting histone-modifying enzymes, such as histone deacetylases (HDAC), has been a widely explored strategy for anticancer drug discovery. The current review focusses on novel aspects of HDAC inhibiting compounds including PROTACs, dual-targeting compounds as well as nanoparticles.

Consequently, I believe that a review article on the topic is timely. Overall the manuscript represents a contribution worthy of publication as review in Molecules. In tandem, I have a few minor concerns about the manuscript but find that it meets the standards associated with publication in Molecules.

Minor comments.

  • Line 256: Roatsh et al. The correct author name is Roatsch
  • Line 262-64: The reported peptides were tested only on cell-lysates and not on intact cells due to the low penetration of the peptide-PROTACs. This should be included here.
  • Line 487: Polimer should be corrected to Polymer

Author Response

Dear Reviewer,:

 Thank you for your nice comments and suggestions. Below you can find a point-by-point answer to your questions

1 -"Line 256: Roatsh et al. The correct author name is Roatsch

We have changed it

2- "Line 262-64: The reported peptides were tested only on cell-lysates and not on intact cells due to the low penetration of the peptide-PROTACs. This should be included here."

The sentence “when tested in cell lysates” has been introduced.

3- "Line 487: Polimer should be corrected to Polymer"

Changed.

Kind Regards

Ana Ramos

Reviewer 2 Report

The presented manuscript entitled "HDAC inhibitors: innovative strategies for their design and application" deals with an interesting topic.

Histone deacetylases (HDACs) are metalloenzymes involved in many gene processes and associated with various disorders, especially cancer. Consequently, many HDAC inhibitors have been discovered in the past and in recent years. The results of these studies have been reported in a number of original papers, and data has been collected in several recent review articles.

The authors of this manuscript try to present the results of research on HDAC inhibitors in a different way.

The strength of the manuscript is the inclusion of very up-to-date data, recently published articles. I suggest to revise this review article.

General comment but for many Figures.

For example Figure 5. In my opinion among compounds 7a – 7d and 8a – 8d should be presented only the most potent one 7c and 8b. The detailed structure-activity-relationships were presented in the original papers. It will also be better to show activity data such as IC50 value or DC50 value in the drawing near the compound structure. This presentation helps you compare the data for different compounds. A detailed description of the activity can be left in the text.

Figure 6. compound 9a instead 9a- 9d.

Figure 8, 9, 12.

Figure 16 -delete.

Figure 25 – introduce IC50 values.

Figure 28 – compound 83 b instead of 83a - 83d.

Please correct the numeration of the Figures in the text – page 24.

Figure 28 – compounds 85a-c with IC50 values.

Figure 29 – compound 88d instead of 88a -e and with IC50 values.

Figure 30, 31,32,33,34 - Same comments as above, compounds with IC50 values.

Author Response

Dear Reviewer,

We acknowledge your nice comments and your suggestions related to the figures, which for sure will improve the presentation of the data.

Below you can find a point-by-point response to your questions:

1- "For example, Figure 5. In my opinion among compounds 7a – 7d and 8a – 8d should be presented only the most potent one 7c and 8b. The detailed structure-activity-relationships were presented in the original papers. It will also be better to show activity data such as IC50 value or DC50 value in the drawing near the compound structure. This presentation helps you compare the data for different compounds. A detailed description of the activity can be left in the text."

We agree with your suggestion. We have left the most active compound in the scheme and added the activity values.

2- "Figure 6. compound 9a instead 9a- 9d."

We have made the suggested change

3- "Figure 8, 9, 12."

We have made the suggested changes except for compounds 13a-d in figure 8, because we discuss in the main text the differences in activity of these derivatives, so it is necessary to maintain them in the figure for the reader to follow the discussion.

4- Figure 16 -delete.

Figure 16 has been deleted, and the numeration of other figures and compounds changed

5- Figure 25 – introduce IC50 values.

This figure (24 after changing the numeration) is about probes which are included in older reviews and we do not describe their activity in the text. Therefore, in our opinion it is not necessary to include their IC50 values in the figure.

6- Figure 28 – compound 83 b instead of 83a - 83d.

The suggested change in the figure 28 has been done and IC50 values have been introduced

7- Please correct the numeration of the Figures in the text – page 24.

We have checked the numeration of the figures in page 24

8- Figure 28 – compounds 85a-c with IC50 values.

This has been done

9- Figure 29 – compound 88d instead of 88a -e and with IC50 values.

This has been done

10- Figure 30, 31,32,33,34 - Same comments as above, compounds with IC50 values.

This has been done

Kind Regards,

Ana Ramos